# A GENERALIZED TRAINING APPROACH FOR MULTIAGENT LEARNING

**Paul Muller**
pmuller@...

**Shayegan Omidshafiei**
somidshafiei@...

**Mark Rowland**
markrowland@...

**Karl Tuyls**
karltuyls@...

**Julien Perolat**
perolat@...

**Siqi Liu**
liusiqi@...

**Daniel Hennes**
hennes@...

**Luke Marris**
marris@...

**Marc Lanctot**
lanctot@...

**Edward Hughes**
edwardhughes@...

**Zhe Wang**
zhewang@...

**Guy Lever**
guylever@...

**Nicolas Heess**
heess@...

**Thore Graepel**
thore@...

**Remi Munos**
munos@...

...google.com.      DeepMind.

## ABSTRACT

This paper investigates a population-based training regime based on game-theoretic principles called Policy-Spaced Response Oracles (PSRO). PSRO is general in the sense that it (1) encompasses well-known algorithms such as fictitious play and double oracle as special cases, and (2) in principle applies to general-sum, many-player games. Despite this, prior studies of PSRO have been focused on two-player zero-sum games, a regime wherein Nash equilibria are tractably computable. In moving from two-player zero-sum games to more general settings, computation of Nash equilibria quickly becomes infeasible. Here, we extend the theoretical underpinnings of PSRO by considering an alternative solution concept, $\alpha$-Rank, which is unique (thus faces no equilibrium selection issues, unlike Nash) and applies readily to general-sum, many-player settings. We establish convergence guarantees in several games classes, and identify links between Nash equilibria and $\alpha$-Rank. We demonstrate the competitive performance of $\alpha$-Rank-based PSRO against an exact Nash solver-based PSRO in 2-player Kuhn and Leduc Poker. We then go beyond the reach of prior PSRO applications by considering 3- to 5-player poker games, yielding instances where $\alpha$-Rank achieves faster convergence than approximate Nash solvers, thus establishing it as a favorable general games solver. We also carry out an initial empirical validation in MuJoCo soccer, illustrating the feasibility of the proposed approach in another complex domain.

## 1 INTRODUCTION

Creating agents that learn to interact in large-scale systems is a key challenge in artificial intelligence. Impressive results have been recently achieved in restricted settings (e.g., zero-sum, two-player games) using game-theoretic principles such as iterative best response computation (Lanctot et al., 2017), self-play (Silver et al., 2018), and evolution-based training (Jaderberg et al., 2019; Liu et al., 2019). A key principle underlying these approaches is to iteratively train a growing population of player policies, with population evolution informed by heuristic skill ratings (e.g., Elo (Elo, 1978)) or game-theoretic solution concepts such as Nash equilibria. A general application of this principle is embodied by the Policy-Space Response Oracles (PSRO) algorithm and its related extensions (Lanctot et al., 2017; Balduzzi et al., 2019). Given a game (e.g., poker), PSRO constructs a higher-level meta-game by simulating outcomes for all match-ups of a population of players' policies. It then trains new policies for each player (via an *oracle*) against a distribution over the existing meta-game

policies (typically an approximate Nash equilibrium, obtained via a *meta-solver*[1]), appends these new policies to the meta-game population, and iterates. In two-player zero sum games, fictitious play (Brown, 1951), double oracle (McMahan et al., 2003), and independent reinforcement learning can all be considered instances of PSRO, demonstrating its representative power (Lanctot et al., 2017).

Prior applications of PSRO have used Nash equilibria as the policy-selection distribution (Lanctot et al., 2017; Balduzzi et al., 2019), which limits the scalability of PSRO to general games: Nash equilibria are intractable to compute in general (Daskalakis et al., 2009); computing *approximate* Nash equilibria is also intractable, even for some classes of two-player games (Daskalakis, 2013); finally, when they can be computed, Nash equilibria suffer from a selection problem (Harsanyi et al., 1988; Goldberg et al., 2013). It is, thus, evident that the reliance of PSRO on the Nash equilibrium as the driver of population growth is a key limitation, preventing its application to general games. Recent work has proposed a scalable alternative to the Nash equilibrium, called $\alpha$-Rank, which applies readily to general games (Omidshafiei et al., 2019), making it a promising candidate for population-based training. Given that the formal study of PSRO has only been conducted under the restricted settings determined by the limitations of Nash equilibria, establishing its theoretical and empirical behaviors under alternative meta-solvers remains an important and open research problem.

We study several PSRO variants in the context of general-sum, many-player games, providing convergence guarantees in several classes of such games for PSRO instances that use $\alpha$-Rank as a meta-solver. We also establish connections between Nash and $\alpha$-Rank in specific classes of games, and identify links between $\alpha$-Rank and the Projected Replicator Dynamics employed in prior PSRO instances (Lanctot et al., 2017). We develop a new notion of best response that guarantees convergence to the $\alpha$-Rank distribution in several classes of games, verifying this empirically in randomly-generated general-sum games. We conduct empirical evaluations in Kuhn and Leduc Poker, first establishing our approach as a competitive alternative to Nash-based PSRO by focusing on two-player variants of these games that have been investigated in these prior works. We subsequently demonstrate empirical results extending beyond the reach of PSRO with Nash as a meta-solver by evaluating training in 3- to 5-player games. Finally, we conduct preliminary evaluations in MuJoCo soccer (Liu et al., 2019), another complex domain wherein we use reinforcement learning agents as oracles in our proposed PSRO variants, illustrating the feasibility of the approach.

## 2 PRELIMINARIES

**Games** We consider $K$-player games, where each player $k \in [K]$ has a finite set of pure strategies $S^k$. Let $S = \prod_k S^k$ denote the space of pure strategy profiles. Denote by $S^{-k} = \prod_{l \neq k} S^l$ the set of pure strategy profiles excluding those of player $k$. Let $\boldsymbol{M}(s) = (\boldsymbol{M}^1(s), \ldots, \boldsymbol{M}^K(s)) \in \mathbb{R}^K$ denote the vector of expected player payoffs for each $s \in S$. A game is said to be *zero-sum* if $\sum_k \boldsymbol{M}^k(s) = 0$ for all $s \in S$. A game is said to be *symmetric* if all players have identical strategy sets $S^k$, and for any permutation $\rho$, strategy profile $(s^1, \ldots, s^K) \in S$, and index $k \in [K]$, one has $\boldsymbol{M}^k(s^1, \ldots, s^K) = \boldsymbol{M}^{\rho(k)}(s^{\rho(1)}, \ldots, s^{\rho(K)})$. A mixed strategy profile is defined as $\boldsymbol{\pi} \in \Delta_S$, a tuple representing the probability distribution over pure strategy profiles $s \in S$. The expected payoff to player $k$ under a mixed strategy profile $\boldsymbol{\pi}$ is given by $\boldsymbol{M}^k(\boldsymbol{\pi}) = \sum_{s \in S} \boldsymbol{\pi}(s)\boldsymbol{M}^k(s)$.

**Nash Equilibrium (NE)** Given a mixed profile $\boldsymbol{\pi}$, the *best response* for a player $k$ is defined $\text{BR}^k(\boldsymbol{\pi}) = \arg\max_{\boldsymbol{\nu} \in \Delta_{S^k}} [\boldsymbol{M}^k(\boldsymbol{\nu}, \boldsymbol{\pi}^{-k})]$. A factorized mixed profile $\boldsymbol{\pi}(s) = \prod_k \boldsymbol{\pi}^k(s^k)$ is a *Nash equilibrium (NE)* if $\boldsymbol{\pi}^k \in \text{BR}^k(\boldsymbol{\pi})$ for all $k \in [K]$. Define $\text{NASHCONV}(\boldsymbol{\pi}) = \sum_k \boldsymbol{M}^k(\text{BR}^k(\boldsymbol{\pi}), \boldsymbol{\pi}^{-k}) - \boldsymbol{M}^k(\boldsymbol{\pi})$; roughly speaking, this measures "distance" from an NE (Lanctot et al., 2017). In prior PSRO instances (Lanctot et al., 2017), a variant of the replicator dynamics (Taylor and Jonker, 1978; Maynard Smith and Price, 1973), called the Projected Replicator Dynamics (PRD), has been used as an approximate Nash meta-solver (see Appendix E for details on PRD).

**$\alpha$-Rank** While NE exist in all finite games (Nash, 1950), their computation is intractable in general games, and their non-uniqueness leads to an equilibrium-selection problem (Harsanyi et al., 1988; Goldberg et al., 2013). This limits their applicability as the underlying driver of training beyond the two-player, zero-sum regime. Recently, an alternate solution concept called $\alpha$-Rank was proposed by

---

[1]A meta-solver is a method that computes, or approximates, the solution concept that is being deployed.

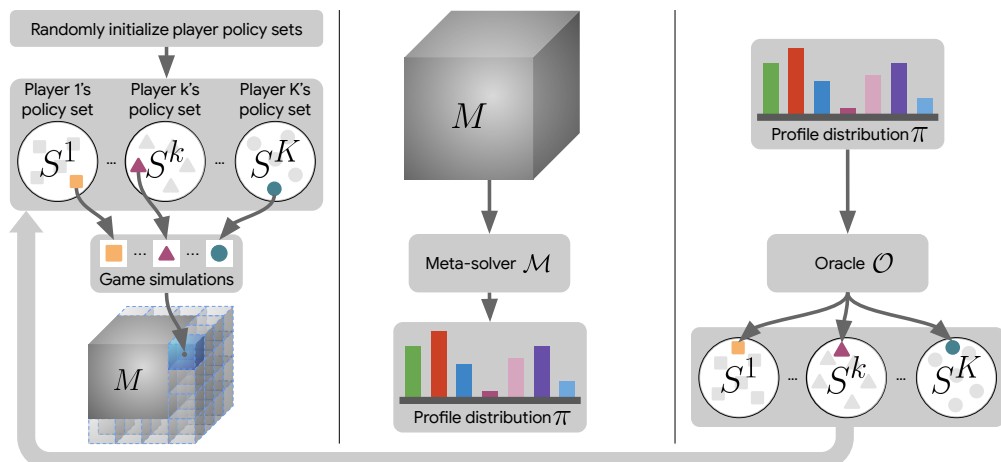

(a) Complete: compute missing payoff tensor $M$ entries via game simulations.

(b) Solve: given the updated payoff tensor $M$, calculate meta-strategy $\pi$ via meta-solver $\mathcal{M}$.

(c) Expand: append a new policy to each player's policy space using the oracle $\mathcal{O}$.

Figure 1: Overview of PSRO($\mathcal{M}$, $\mathcal{O}$) algorithm phases.

Omidshafiei et al. (2019), the key associated benefits being its uniqueness and efficient computation in many-player and general-sum games, making it a promising means for directing multiagent training.

The $\alpha$-Rank distribution is computed by constructing the *response graph* of the game: each strategy profile $s \in S$ of the game is a node of this graph; a directed edge points from any profile $s \in S$ to $\sigma \in S$ in the graph if (1) $s$ and $\sigma$ differ in only a single player $k$'s strategy and (2) $M^k(\sigma) > M^k(s)$. $\alpha$-Rank constructs a random walk along this directed graph, perturbing the process by injecting a small probability of backwards-transitions from $\sigma$ to $s$ (dependent on a parameter, $\alpha$, whose value is prescribed by the algorithm); this ensures irreducibility of the resulting Markov chain and the existence of a unique stationary distribution, $\pi \in \Delta_S$, called the $\alpha$-Rank distribution. The masses of $\pi$ are supported by the sink strongly-connected components (SSCCs) of the response graph (Omidshafiei et al., 2019). For more details on $\alpha$-Rank, see Appendix D and Rowland et al. (2019).

**Oracles** We define an oracle $\mathcal{O}$ as an abstract computational entity that, given a game, computes policies with precise associated properties. For instance, a best-response oracle $\mathcal{O}^k(\pi) = \mathrm{BR}^k(\pi)$ computes the best-response policy for any player $k$, given a profile $\pi$. One may also consider approximate-best-response oracles that, e.g., use reinforcement learning to train a player $k$'s policy against a fixed distribution over the other players' policies, $\pi^{-k}$. Oracles play a key role in population-based training, as they compute the policies that are incrementally added to players' growing policy populations (McMahan et al., 2003; Lanctot et al., 2017; Balduzzi et al., 2019). The choice of oracle $\mathcal{O}$ also affects the training convergence rate and final equilibrium reached (e.g., Nash or $\alpha$-Rank).

**Empirical Game-theoretic Analysis** PSRO relies on principles from empirical game-theoretic analysis (EGTA) (Walsh et al., 2002; Phelps et al., 2004; Wellman, 2006). Given a game (e.g., poker), EGTA operates via construction of a higher-level 'meta-game', where strategies $s$ correspond to policies (e.g., 'play defensively' in poker) rather than atomic actions (e.g., 'fold'). A meta-payoff table $M$ is then constructed by simulating games for all joint policy combinations, with entries corresponding to the players' expected utilities under these policies. Game-theoretic analysis can then be conducted on the meta-game in a manner analogous to the lower-level game, albeit in a much more scalable manner. As the theoretical discussion hereafter pertains to the meta-game, we use $s$, $M$, and $\pi$ to respectively refer to policies, payoffs, and distributions at the meta-level, rather than the underlying low-level game. In our analysis, it will be important to distinguish between SSCCs of the underlying game, and of the meta-game constructed by PSRO; we refer to the latter as meta-SSCCs.

## 3 POLICY-SPACE RESPONSE ORACLES: NASH AND BEYOND

We first overview Policy-Space Response Oracles (PSRO) prior to presenting our findings. Given an underlying game (e.g., Poker), PSRO first initializes the policy space $S$ using randomly-generated policies, then expands the players' policy populations in three iterated phases: **complete**, **solve**, and

---

**Algorithm 1** PSRO($\mathcal{M}, \mathcal{O}$)

---

1: Initialize the players' policy set $S = \prod_k S^k$ via random policies
2: **for** iteration $\in \{1, 2, \cdots\}$ **do**
3:     Update payoff tensor $M$ for new policy profiles in $S$ via game simulations        $\triangleright$ (Fig. 1a)
4:     Compute the meta-strategy $\pi$ using meta-solver $\mathcal{M}(M)$        $\triangleright$ (Fig. 1b)
5:     Expand the policy space for each player $k \in [K]$ via $S^k \leftarrow S^k \cup \mathcal{O}^k(\pi)$        $\triangleright$ (Fig. 1c)

---

| Game type | $\mathcal{M}$ | $\mathcal{O}$ | Converges to $\alpha$-Rank? |
|-----------|---------------|---------------|------------------------------|
| SP | $\alpha$-Rank | BR | ✗ (Example 1) |
| SP | $\alpha$-Rank | PBR | ✓ (Sub-SSCC,[†] Proposition 3) |
| MP | $\alpha$-Rank | BR | ✗ (Example 2) |
| MP | $\alpha$-Rank | PBR | ✓ (With novelty-bound oracle,[†] Proposition 1) |
| SP / MP | Uniform or Nash | BR | ✗ (Examples 4 and 5, Appendix A.2) |

Table 1: Theory overview. SP and MP, resp., denote single and multi-population games. BR and PBR, resp., denote best response and preference-based best response. [†]Defined in the noted propositions.

**expand** (see Algorithm 1 and Fig. 1). In the **complete** phase, a meta-game consisting of all match-ups of these joint policies is synthesized, with missing payoff entries in $M$ completed through game simulations. Next, in the **solve** phase, a meta-solver $\mathcal{M}$ computes a profile $\pi$ over the player policies (e.g., Nash, $\alpha$-Rank, or uniform distributions). Finally, in the **expand** phase, an oracle $\mathcal{O}$ computes at least one new policy $s'_k$ for each player $k \in [K]$, given profile $\pi$. As other players' policy spaces $S^{-k}$ and profile $\pi^{-k}$ are fixed, this phase involves solving a single-player optimization problem. The new policies are appended to the respective players' policy sets, and the algorithm iterates. We use PSRO($\mathcal{M}, \mathcal{O}$) to refer to the PSRO instance using meta-solver $\mathcal{M}$ and oracle $\mathcal{O}$. Notably, PSRO-based training for two-player symmetric games can be conducted using a single population of policies that is shared by all players (i.e., $S^k$ is identical for all $k$). Thus, we henceforth refer to two-player symmetric games as 'single-population games', and more generally refer to games that require player-specific policy populations as 'multi-population games'. Recent investigations of PSRO have solely focused on Nash-based meta-solvers and best-response-based oracles (Lanctot et al., 2017; Balduzzi et al., 2019), with theory focused around the two-player zero-sum case. Unfortunately, these guarantees do not hold in games beyond this regime, making investigation of alternative meta-solvers and oracles critical for further establishing PSRO's generalizability.

## 4 GENERALIZING PSRO THEORY

This section establishes theoretical properties of PSRO for several useful classes of general games. We summarize our results in Table 1, giving a full exposition below.

### 4.1 ESTABLISHING CONVERGENCE TO $\alpha$-RANK

It is well-known that PSRO(Nash, BR) will eventually return an NE in two-player zero-sum games (McMahan et al., 2003). In more general games, where Nash faces the issues outlined earlier, $\alpha$-Rank appears a promising meta-solver candidate as it applies to many-player, general-sum games and has no selection problem. However, open questions remain regarding convergence guarantees of PSRO when using $\alpha$-Rank, and whether standard BR oracles suffice for ensuring these guarantees. We investigate these theoretical questions, namely, whether particular variants of PSRO can converge to the $\alpha$-Rank distribution for the underlying game.

|  |  | Player 2 |  |  |  |
|---|---|---|---|---|---|
|  | $A$ | $B$ | $C$ | $D$ | $X$ |
| $A$ | 0 | $-\phi$ | 1 | $\phi$ | $-\varepsilon$ |
| $B$ | $\phi$ | 0 | $-\phi^2$ | 1 | $-\varepsilon$ |
| $C$ | $-1$ | $\phi^2$ | 0 | $-\phi$ | $-\varepsilon$ |
| $D$ | $-\phi$ | $-1$ | $\phi$ | 0 | $-\varepsilon$ |
| $X$ | $\varepsilon$ | $\varepsilon$ | $\varepsilon$ | $\varepsilon$ | 0 |

(Player 1 labels the rows)

Table 2: Symmetric zero-sum game used to analyze the behavior of PSRO in Example 1. Here, $0 < \varepsilon \ll 1$ and $\phi \gg 1$.

A first attempt to establish convergence to $\alpha$-Rank might involve running PSRO to convergence (until the oracle returns a strategy already in the convex hull of the known strategies), using $\alpha$-Rank as the meta-solver, and a standard best response oracle. However, the following example shows that this will not work in general for the single-population case (see Fig. A.5 for a step-by-step illustration).

**Example 1.** *Consider the symmetric zero-sum game specified in Table 2. As $X$ is the sole sink component of the game's response graph (as illustrated in Fig. A.5a), the single-population $\alpha$-Rank distribution for this game puts unit mass on $X$. We now show that a PSRO algorithm that computes best responses to the $\alpha$-Rank distribution over the current strategy set need not recover strategy $X$, by computing directly the strategy sets of the algorithm initialized with the set $\{C\}$.*

1. *The initial strategy space consists only of the strategy $C$; the best response against $C$ is $D$.*
2. *The $\alpha$-Rank distribution over $\{C, D\}$ puts all mass on $D$; the best response against $D$ is $A$.*
3. *The $\alpha$-Rank distribution over $\{C, D, A\}$ puts all mass on $A$; the best response against $A$ is $B$.*
4. *The $\alpha$-Rank distribution over $\{C, D, A, B\}$ puts mass $(1/3, 1/3, 1/6, 1/6)$ on $(A, B, C, D)$ respectively. For $\phi$ sufficiently large, the payoff that $C$ receives against $B$ dominates all others, and since $B$ has higher mass than $C$ in the $\alpha$-Rank distribution, the best response is $C$.*

*Thus, PSRO($\alpha$-Rank, BR) leads to the algorithm terminating with strategy set $\{A, B, C, D\}$ and not discovering strategy $X$ in the sink strongly-connected component.*

This conclusion also holds in the multi-population case, as the following counterexample shows.

**Example 2.** *Consider the game in Table 2, treating it now as a multi-population problem. It is readily verified that the multi-population $\alpha$-Rank distributions obtained by PSRO with initial strategy sets consisting solely of $C$ for each player are: (i) a Dirac delta at the joint strategy $(C, C)$, leading to best responses of $D$ for both players; (ii) a Dirac delta at $(D, D)$ leading to best responses of $A$ for both players; (iii) a Dirac delta at $(A, A)$, leading to best responses of $B$ for both players; and finally (iv) a distribution over joint strategies of the 4×4 subgame induced by strategies $A, B, C, D$ that leads to a best response* not *equal to $X$; thus, the full $\alpha$-Rank distribution is again* not *recovered.*

### 4.2 A New Response Oracle

The previous examples indicate that the use of standard best responses in PSRO may be the root cause of the incompatibility with the $\alpha$-Rank solution concept. Thus, we define the *Preference-based Best Response (PBR) oracle*, which is more closely aligned with the dynamics defining $\alpha$-Rank, and which enables us to establish desired PSRO guarantees with respect to $\alpha$-Rank.

Consider first the single-population case. Given an $N$-strategy population $\{s_1, \ldots, s_N\}$ and corresponding meta-solver distribution $(\boldsymbol{\pi}_i)_{i=1}^N \in \Delta_N$, a PBR oracle is defined as any function satisfying

$$\mathrm{PBR}\left(\sum_i \boldsymbol{\pi}_i s_i\right) \subseteq \arg\max_\sigma \sum_i \boldsymbol{\pi}_i \mathbb{1}\left[\boldsymbol{M}^1(\sigma, s_i) > \boldsymbol{M}^2(\sigma, s_i)\right], \tag{1}$$

where the $\arg\max$ returns the *set* of policies optimizing the objective, and the optimization is over pure strategies in the underlying game. The intuition for the definition of PBR is that we would like the oracle to return strategies that will receive high mass under $\alpha$-Rank when added to the population; objective (1) essentially encodes the probability flux that the vertex corresponding to $\sigma$ would receive in the random walk over the $\alpha$-Rank response graph (see Section 2 or Appendix D for further details).

We demonstrate below that the use of the PBR resolves the issue highlighted in Example 1 (see Fig. A.6 in Appendix A for an accompanying visual).

**Example 3.** *Steps 1 to 3 of correspond exactly to those of Example 1. In step 4, the $\alpha$-Rank distribution over $\{C, D, A, B\}$ puts mass $(1/3, 1/3, 1/6, 1/6)$ on $(A, B, C, D)$ respectively. A beats $C$ and $D$, thus its PBR score is $1/3$. B beats $A$ and $D$, thus its PBR score is $1/2$. C beats $B$, its PBR score is thus $1/3$. D beats $C$, its PBR score is thus $1/6$. Finally, $X$ beats every other strategy, and its PBR score is thus $1$. Thus, there is only one strategy maximizing PBR, $X$, which is then chosen, thereby recovering the SSCC of the game and the correct $\alpha$-Rank distribution at the next timestep.*

In the multi-population case, consider a population of $N$ strategy profiles $\{s_1, \ldots, s_N\}$ and corresponding meta-solver distribution $(\boldsymbol{\pi}_i)_{i=1}^N$. Several meta-SSCCs may exist in the multi-population $\alpha$-Rank response graph. In this case, we run the PBR oracle for each meta-SSCC separately, as follows. Suppose there are $\ell$ meta-SSCCs, and denote by $\boldsymbol{\pi}^{(\ell)}$ the distribution $\boldsymbol{\pi}$ restricted to the $\ell^{\mathrm{th}}$ meta-SSCC, for all $1 \leq \ell \leq L$. The PBR for player $k$ on the $\ell^{\mathrm{th}}$ meta-SSCC is then defined by

$$\mathrm{PBR}^k\left(\sum_i \boldsymbol{\pi}_i^{(\ell)} s_i\right) \subseteq \arg\max_\sigma \sum_i \boldsymbol{\pi}_i^{(\ell)} \mathbb{1}\left[\boldsymbol{M}^k(\sigma, s_i^{-k}) > \boldsymbol{M}^k(s_i^k, s_i^{-k})\right]. \tag{2}$$

Thus, the PBR oracle generates one new strategy for each player for every meta-SSCC in the $\alpha$-Rank response graph; we return this full set of strategies and append to the policy space accordingly, as

in Line 5 of Algorithm 1. Intuitively, this leads to a *diversification* of strategies introduced by the oracle, as each new strategy need only perform well against a subset of prior strategies. This hints at interesting links with the recently-introduced concept of rectified-Nash BR (Balduzzi et al., 2019), which also attempts to improve diversity in PSRO, albeit only in two-player zero-sum games.

We henceforth denote PSRO($\alpha$-Rank, PBR) as $\alpha$-PSRO for brevity. We next define $\alpha$-CONV, an approximate measure of convergence to $\alpha$-Rank. We restrict discussion to the multi-population case here, describing the single-population case in Appendix A.4. With the notation introduced above, we define PBR-SCORE$^k(\sigma; \boldsymbol{\pi}, S) = \sum_i \sum_\ell \boldsymbol{\pi}_i^{(\ell)} \mathbb{1} \left[ \boldsymbol{M}^k(\sigma, s_i^{-k}) > \boldsymbol{M}^k(s_i^k, s_i^{-k}) \right]$, and

$$\alpha\text{-CONV} = \sum_k \max_\sigma \text{PBR-SCORE}^k(\sigma) - \max_{s \in S^k} \text{PBR-SCORE}^k(s),$$

where $\max_\sigma$ is taken over the pure strategies of the underlying game. Unfortunately, in the multi-population case, a PBR-SCORE of 0 does not necessarily imply $\alpha$-partial convergence. We thus introduce a further measure, PCS-SCORE, defined by PCS-SCORE $= \frac{\text{\# of } \alpha\text{-PSRO strategy profiles in the underlying game's SSCCs}}{\text{\# of } \alpha\text{-PSRO strategy profiles in meta-SSCCs}}$, which assesses the quality of the $\alpha$-PSRO population. We refer readers to Appendix C.3 for pseudocode detailing how to implement these measures in practice.

### 4.3  $\alpha$-PSRO: THEORY, PRACTICE, AND CONNECTIONS TO NASH

We next study the theoretical properties of PSRO($\alpha$-Rank, PBR), which we henceforth refer to as $\alpha$-PSRO for brevity. We consider that $\alpha$-PSRO has converged if no new strategy has been returned by PBR for any player at the end of an iteration. Proofs of all results are provided in Appendix B.

**Definition 1.** *A PSRO algorithm is said to converge $\alpha$-**fully** (resp., $\alpha$-**partially**) to an SSCC of the underlying game if its strategy population contains the full SSCC (resp., a sub-cycle of the SSCC, denoted a 'sub-SSCC') after convergence.*

**Definition 2.** *We also adapt PBR to be what we call **novelty-bound** by restricting the $\arg\max$ in Equation* (1) *to be over strategies not already included in the population with* PBR-SCORE $> 0$.

In particular, the novelty-bound version of the PBR oracle is given by restricting the arg max appearing in (2) to only be over strategies not already present in the population.

These definitions enable the following results for $\alpha$-PSRO in the single- and multi-population cases.

**Proposition 1.** *If at any point the population of $\alpha$-PSRO contains a member of an SSCC of the game, then $\alpha$-PSRO will $\alpha$-partially converge to that SSCC.*

**Proposition 2.** *If we constrain the PBR oracle used in $\alpha$-PSRO to be novelty-bound, then $\alpha$-PSRO will $\alpha$-fully converge to at least one SSCC of the game.*

Stronger guarantees exist for two-players symmetric (i.e., single-population) games, though the multi-population case encounters more issues, as follows.

**Proposition 3.** *(Single-population) $\alpha$-PSRO converges $\alpha$-partially to the unique SSCC.*

**Proposition 4.** *(Multi-population) Without a novelty-bound oracle, there exist games for which $\alpha$-PSRO does not converge $\alpha$-partially to any SSCC.*

Intuitively, the lack of convergence without a novelty-bound oracle can occur due to intransitivities in the game (i.e., cycles in the game can otherwise trap the oracle). An example demonstrating this issue is shown in Fig. B.7, with an accompanying step-by-step walkthrough in Appendix B.4. Specifically, SSCCs may be hidden by "intermediate" strategies that, while not receiving as high a payoff as current population-pool members, can actually lead to well-performing strategies outside the population. As these "intermediate" strategies are avoided, SSCCs are consequently not found. Note also that this is related to the common problem of action/equilibrium shadowing, as detailed in Matignon et al. (2012).

In Section 5, we further investigate convergence behavior beyond the conditions studied above. In practice, we demonstrate that despite the negative result of Proposition 4, $\alpha$-PSRO does significantly increase the probability of converging to an SSCC, in contrast to PSRO(Nash, BR). Overall, we have shown that for general-sum multi-player games, it is possible to give theoretical guarantees

for a version of PSRO driven by $\alpha$-Rank in several circumstances. By contrast, using exact NE in PSRO is intractable in general. In prior work, this motivated the use of approximate Nash solvers generally based on the simulation of dynamical systems or regret minimization algorithms, both of which generally require specification of several hyperparameters (e.g., simulation iterations, window sizes for computing time-average policies, and entropy-injection rates), and a greater computational burden than $\alpha$-Rank to carry out the simulation in the first place.

**Implementing the PBR Oracle**  Recall from Section 3 that the BR oracle inherently solves a single-player optimization problem, permitting use of a single-agent RL algorithm as a BR approximator, which is useful in practice. As noted in Section 4.1, however, there exist games where the BR and PBR objectives are seemingly incompatible, preventing the use of standard RL agents for PBR approximation. While exact PBR is computable in small-scale (e.g., normal-form) games, we next consider more general games classes where PBR can also be approximated using standard RL agents.

**Definition 3.** *Objective $\mathcal{A}$ is 'compatible' with objective $\mathcal{B}$ if any solution to $\mathcal{A}$ is a solution to $\mathcal{B}$.*

**Proposition 5.** *A constant-sum game is denoted as **win-loss** if $M^k(s) \in \{0, 1\}$ for all $k \in [K]$ and $s \in S$. BR is compatible with PBR in win-loss games in the two-player single-population case.*

**Proposition 6.** *A symmetric two-player game is denoted **monotonic** if there exists a function $f : S \to \mathbb{R}$ and a non-decreasing function $\sigma : \mathbb{R} \to \mathbb{R}$ such that $M^1(s, \nu) = \sigma(f(s) - f(\nu))$. BR is compatible with PBR in monotonic games in the single-population case.*

Finally, we next demonstrate that under certain conditions, there are strong connections between the PBR objective defined above and the broader field of preference-based RL (Wirth et al., 2017).

**Proposition 7.** *Consider symmetric win-loss games where outcomes between deterministic strategies are deterministic. A preference-based RL agent (i.e., an agent aiming to maximize its probability of winning against a distribution $\boldsymbol{\pi}$ of strategies $\{s_1, \ldots, s_N\}$) optimizes exactly the PBR objective* (1).

Given this insight, we believe an important subject of future work will involve the use of preference-based RL algorithms in implementing the PBR oracle for more general classes of games. We conclude this section with some indicative results of the relationship between $\alpha$-Rank and NE.

**Proposition 8.** *For symmetric two-player zero-sum games where off-diagonal payoffs have equal magnitude, all NE have support contained within that of the single-population $\alpha$-Rank distribution.*

**Proposition 9.** *In a symmetric two-player zero-sum game, there exists an NE with support contained within that of the $\alpha$-Rank distribution.*

For more general games, the link between $\alpha$-Rank and Nash equilibria will likely require a more complex description. We leave this for future work, providing additional discussion in Appendix A.3.

## 5 EVALUATION

We conduct evaluations on games of increasing complexity, extending beyond prior PSRO applications that have focused on two-player zero-sum games. For experimental procedures, see Appendix C.

**Oracle comparisons**  We evaluate here the performance of the BR and PBR oracles in games where PBR can be exactly computed. We consider randomly generated, $K$-player, general-sum games with increasing strategy space sizes, $|S^k|$. Figure 2 reports these results for the 4- and 5-player instances (see Appendix C.4 for 2-3 player results). The asymmetric nature of these games, in combination with the number of players and strategies involved, makes them inherently, and perhaps surprisingly, large in scale. For example, the largest considered game in Fig. 2 involves 5 players with 30 strategies each, making for a total of more than 24 million strategy profiles in total. For each combination of $K$ and $|S^k|$, we generate $1e6$ random games. We conduct 10 trials per game, in each trial running the BR and PBR oracles starting from a random strategy in the corresponding response graph, then iteratively expanding the population space until convergence. Importantly, this implies that the starting strategy may not even be in an SSCC.

As mentioned in Section 4.2, $\alpha$-CONV and PCS-SCORE jointly characterize the oracle behaviors in these multi-population settings. Figure 2a plots $\alpha$-CONV for both oracles, demonstrating that PBR outperforms BR in the sense that it captures more of the game SSCCs. Figures 2b and 2c, respectively,

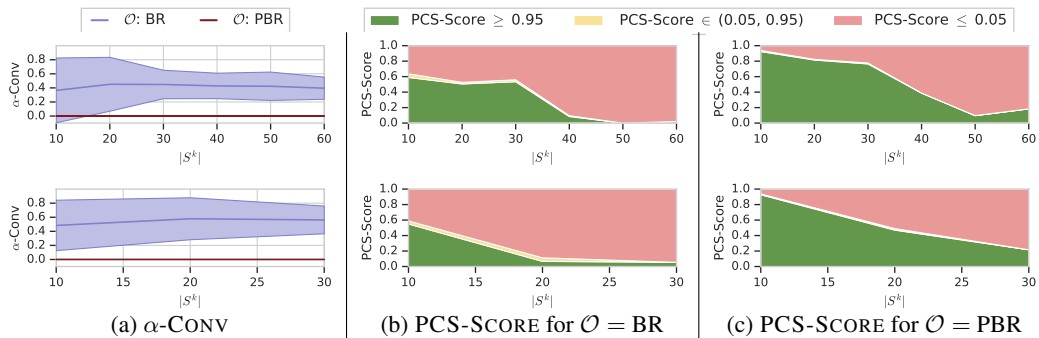

Figure 2: Oracle comparisons for randomly-generated games with varying player strategy space sizes $|S^k|$. Top and bottom rows, respectively, correspond to 4- and 5-player games.

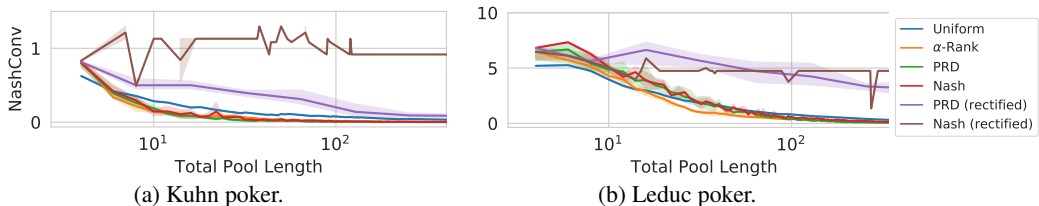

Figure 3: Results for 2-player poker domains.

plot the PCS-SCORE for BR and PBR over all game instances. The PCS-SCORE here is typically either (a) greater than 95%, or (b) less than 5%, and otherwise rarely between 5% to 95%. For all values of $|S^k|$, PBR consistently discovers a larger proportion of the $\alpha$-Rank support in contrast to BR, serving as useful validation of the theoretical results of Section 4.3.

**Meta-solver comparisons** We consider next the standard benchmarks of Kuhn and Leduc poker (Kuhn, 1950; Southey et al., 2005; Lanctot et al., 2019). We detail these domains in Appendix C.2, noting here that both are $K$-player, although Leduc is significantly more complex than Kuhn. We first consider two-player instances of these poker domains, permitting use of an exact Nash meta-solver. Figure 3 compares the NASHCONV of PSRO($\mathcal{M}$, BR) for various meta-solver $\mathcal{M}$ choices. Note that the x axis of Figure 3 and Figure 4 is the Total Pool Length (The sum of the length of each player's pool in PSRO) instead of the number of iterations of PSRO, since Rectified solvers can add more than one policy to the pool at each PSRO iteration (Possibly doubling pool size at every PSRO iteration). It is therefore more pertinent to compare exploitabilities at the same pool sizes rather than at the same number of PSRO iterations.

In Kuhn poker (Fig. 3a), the $\alpha$-Rank, Nash, and the Projected Replicator Dynamics (PRD) meta-solvers converge essentially at the same rate towards zero NASHCONV, in contrast to the slower rate of the Uniform meta-solver, the very slow rate of the Rectified PRD solver, and the seemingly constant NASHCONV of the Rectified Nash solver. We provide in Appendix C.5 a walkthrough of the first steps of the Rectified Nash results to more precisely determine the cause of its plateauing NASHCONV. A high level explanation thereof is that it is caused by Rectified Nash cycling through the same policies, effectively not discovering new policies. We posit these characteristics, antipodal to the motivation behind Rectified Nash, come from the important fact that Rectified Nash was designed to work only in symmetric games, and is therefore not inherently well-suited for the Kuhn and Leduc poker domains investigated here, as they are both asymmetric games. We did not add the Rectified PRD results the other, greater-than-2 players experiments, as its performance remained non-competitive.

As noted in Lanctot et al. (2017), PSRO(Uniform, BR) corresponds to Fictitious Play (Brown, 1951) and is thus guaranteed to find an NE in such instances of two-player zero-sum games. Its slower convergence rate is explained by the assignment of uniform mass across all policies $s \in S$, implying that PSRO essentially wastes resources on training the oracle to beat even poor-performing strategies. While $\alpha$-Rank does not seek to find an approximation of Nash, it nonetheless reduces the NASHCONV yielding competitive results in comparison to an exact-Nash solver in these instances. Notably, the similar performance of $\alpha$-Rank and Nash serves as empirical evidence that $\alpha$-Rank can be applied

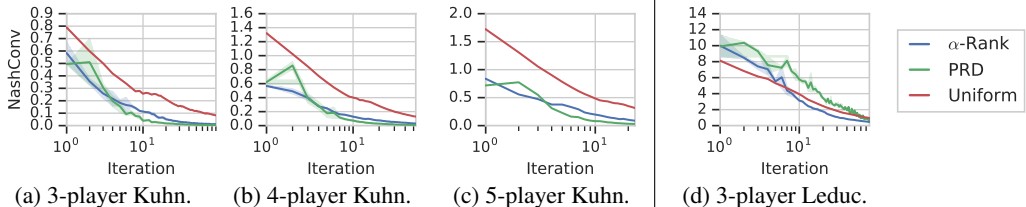

Figure 4: Results for poker domains with more than 2 players.

competitively even in the two-player zero-sum setting, while also showing great promise to be deployed in broader settings where Nash is no longer tractable.

We next consider significantly larger variants of Kuhn and Leduc Poker involving more than two players, extending beyond the reach of prior PSRO results (Lanctot et al., 2017). Figure 4 visualizes the NASHCONV of PSRO using the various meta-solvers (with the exception of an exact Nash solver, due to its intractability in these instances). In all instances of Kuhn Poker, $\alpha$-Rank and PRD show competitive convergence rates. In 3-player Leduc poker, however, $\alpha$-Rank shows fastest convergence, with Uniform following throughout most of training and PRD eventually reaching a similar NASHCONV. Several key insights can be made here. First, computation of an approximate Nash via PRD involves simulation of the associated replicator dynamics, which can be chaotic (Palaiopanos et al., 2017) even in two-player two-strategy games, making it challenging to determine when PRD has suitably converged. Second, the addition of the projection step in PRD severs its connection with NE; the theoretical properties of PRD were left open in Lanctot et al. (2017), leaving it without any guarantees. These limitations go beyond theoretical, manifesting in practice, e.g., in Fig. 4d, where PRD is outperformed by even the uniform meta-solver for many iterations. Given these issues, we take a first (and informal) step towards analyzing PRD in Appendix E. For $\alpha$-Rank, by contrast, we both establish theoretical properties in Section 4, and face no simulation-related challenges as its computation involves solving of a linear system, even in the general-sum many-player case (Omidshafiei et al., 2019), thus establishing it as a favorable and general PSRO meta-solver.

**MuJoCo Soccer** While the key objective of this paper is to take a first step in establishing a theoretically-grounded framework for PSRO-based training of agents in many-player settings, an exciting question regards the behaviors of the proposed $\alpha$-Rank-based PSRO algorithm in complex domains where function-approximation-based policies need to be relied upon. In Appendix F, we take a first step towards conducting this investigation in the MuJoCo soccer domain introduced in Liu et al. (2019). We remark that these results, albeit interesting, are primarily intended to lay the foundation for use of $\alpha$-Rank as a meta-solver in complex many agent domains where RL agents serve as useful oracles, warranting additional research and analysis to make conclusive insights.

## 6 RELATED WORK

We discuss the most closely related work along two axes. We start with PSRO-based research and some multiagent deep RL work that focuses on training of networks in various multiagent settings. Then we continue with related work that uses evolutionary dynamics ($\alpha$-Rank and replicator dynamics) as a solution concept to examine underlying behavior of multiagent interactions using meta-games.

Policy-space response oracles (Lanctot et al., 2017) unify many existing approaches to multiagent learning. Notable examples include fictitious play (Brown, 1951; Robinson, 1951), independent reinforcement learning (Matignon et al., 2012) and the double oracle algorithm (McMahan et al., 2003). PSRO also relies, fundamentally, on principles from empirical game-theoretic analysis (EGTA) (Walsh et al., 2002; Phelps et al., 2004; Tuyls et al., 2018; Wellman, 2006; Vorobeychik, 2010; Wiedenbeck and Wellman, 2012; Wiedenbeck et al., 2014). The related Parallel Nash Memory (PNM) algorithm (Oliehoek et al., 2006), which can also be seen as a generalization of the double oracle algorithm, incrementally grows the space of strategies, though using a search heuristic rather than exact best responses. PNMs have been successfully applied to games settings utilizing function approximation, notably to address exploitability issues when training Generative Adversarial Networks (GANs) (Oliehoek et al., 2019).

PSRO allows the multiagent learning problem to be decomposed into a sequence of single-agent learning problems. A wide variety of other approaches that deal with the multiagent learning problem without this reduction are also available, such as Multiagent Deep Deterministic Policy Gradients (MADDPG) (Lowe et al., 2017), Counterfactual Multiagent Policy Gradients (COMA) (Foerster et al., 2018), Differentiable Inter-Agent Learning (DIAL) (Foerster et al., 2016), Hysteretic Deep Recurrent Q-learning (Omidshafiei et al., 2017), and lenient Multiagent Deep Reinforcement Learning (Palmer et al., 2018). Several notable contributions have also been made in addressing multiagent learning challenges in continuous-control settings, most recently including the approaches of Iqbal and Sha (2019); Gupta et al. (2017); Wei et al. (2018); Peng et al. (2017); Khadka et al. (2019). We refer interested readers to the following survey of recent deep multiagent RL approaches Hernandez-Leal et al. (2019).

$\alpha$-Rank was introduced by Omidshafiei et al. (2019) as a scalable dynamic alternative to Nash equilibria that can be applied in general-sum, many-player games and is capable of capturing the underlying multiagent evolutionary dynamics. Concepts from evolutionary dynamics have long been used in analysis of multiagent interactions from a meta-game standpoint (Walsh et al., 2002; Tuyls and Parsons, 2007; Hennes et al., 2013; Bloembergen et al., 2015; Tuyls et al., 2018).

## 7 DISCUSSION

This paper studied variants of PSRO using $\alpha$-Rank as a meta-solver, which were shown to be competitive with Nash-based PSRO in zero-sum games, and scale effortlessly to general-sum many-player games, in contrast to Nash-based PSRO. We believe there are many interesting directions for future work, including how uncertainty in the meta-solver distribution, informed by recent developments in dealing with incomplete information in games (Reeves and Wellman, 2004; Walsh et al., 2003; Rowland et al., 2019), can be used to inform the selection of new strategies to be added to populations. In summary, we strongly believe that the theoretical and empirical results established in this paper will play a key role in scaling up multiagent training in general settings.

## ACKNOLWEDGEMENTS

The authors gratefully thank Bart De Vylder for providing helpful feedback on the paper draft.

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

# APPENDICES

## A  EXAMPLES

### A.1  FURTHER EXPOSITION OF EXAMPLES 1 AND 2

Player 2

|  | | A | B | C | D | X |
|---|---|---|---|---|---|---|
| | A | 0 | $-\phi$ | 1 | $\phi$ | $-\varepsilon$ |
| | B | $\phi$ | 0 | $-\phi^2$ | 1 | $-\varepsilon$ |
| Player 1 | C | $-1$ | $\phi^2$ | 0 | $-\phi$ | $-\varepsilon$ |
| | D | $-\phi$ | $-1$ | $\phi$ | 0 | $-\varepsilon$ |
| | X | $\varepsilon$ | $\varepsilon$ | $\varepsilon$ | $\varepsilon$ | 0 |

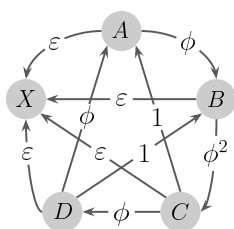

(a) Overview. Full payoff table on left, full response graph on right, with values over directed edges indicating the payoff gained by deviating from one strategy to another.

Player 2

|  | | A | B | C | D | X |
|---|---|---|---|---|---|---|
| | A | 0 | $-\phi$ | 1 | $\phi$ | $-\varepsilon$ |
| | B | $\phi$ | 0 | $-\phi^2$ | 1 | $-\varepsilon$ |
| Player 1 | C | $-1$ | $\phi^2$ | 0 | $-\phi$ | $-\varepsilon$ |
| | D | $-\phi$ | $-1$ | $\boldsymbol{\phi}$ | 0 | $-\varepsilon$ |
| | X | $\varepsilon$ | $\varepsilon$ | $\varepsilon$ | $\varepsilon$ | 0 |

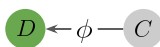

(b) Consider an initial strategy space consisting only of the strategy $C$; the best response against $C$ is $D$.

Player 2

|  | | A | B | C | D | X |
|---|---|---|---|---|---|---|
| | A | 0 | $-\phi$ | 1 | $\boldsymbol{\phi}$ | $-\varepsilon$ |
| | B | $\phi$ | 0 | $-\phi^2$ | 1 | $-\varepsilon$ |
| Player 1 | C | $-1$ | $\phi^2$ | 0 | $-\phi$ | $-\varepsilon$ |
| | D | $-\phi$ | $-1$ | $\phi$ | 0 | $-\varepsilon$ |
| | X | $\varepsilon$ | $\varepsilon$ | $\varepsilon$ | $\varepsilon$ | 0 |

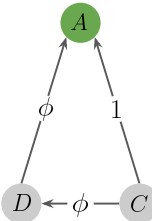

(c) The $\alpha$-Rank distribution over $\{C, D\}$ puts all mass on $D$; the best response against $D$ is $A$.

Player 2

|  | | A | B | C | D | X |
|---|---|---|---|---|---|---|
| | A | 0 | $-\phi$ | 1 | $\phi$ | $-\varepsilon$ |
| | B | $\boldsymbol{\phi}$ | 0 | $-\phi^2$ | 1 | $-\varepsilon$ |
| Player 1 | C | $-1$ | $\phi^2$ | 0 | $-\phi$ | $-\varepsilon$ |
| | D | $-\phi$ | $-1$ | $\phi$ | 0 | $-\varepsilon$ |
| | X | $\varepsilon$ | $\varepsilon$ | $\varepsilon$ | $\varepsilon$ | 0 |

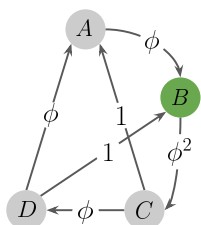

(d) The $\alpha$-Rank distribution over $\{C, D, A\}$ puts all mass on $A$; the best response against $A$ is $B$.

Player 2

|  | | A | B | C | D | X |
|---|---|---|---|---|---|---|
| | A | 0 | $-\phi$ | 1 | $\phi$ | $-\varepsilon$ |
| | B | $\phi$ | 0 | $-\phi^2$ | 1 | $-\varepsilon$ |
| Player 1 | C | $-1$ | $\boldsymbol{\phi^2}$ | 0 | $-\phi$ | $-\varepsilon$ |
| | D | $-\phi$ | $-1$ | $\phi$ | 0 | $-\varepsilon$ |
| | X | $\varepsilon$ | $\varepsilon$ | $\varepsilon$ | $\varepsilon$ | 0 |

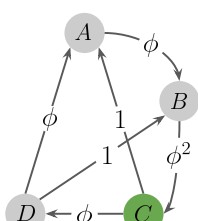

(e) The $\alpha$-Rank distribution over $\{C, D, A, B\}$ puts mass $(1/3, 1/3, 1/6, 1/6)$ on $(A, B, C, D)$ respectively. For $\phi$ sufficiently large, the payoff that $C$ receives against $B$ dominates all others, and since $B$ has higher mass than $C$ in the $\alpha$-Rank distribution, the best response is $C$.

Figure A.5: Example 1 with oracle $\mathcal{O} = \text{BR}$. In each step above, the $\alpha$-Rank support is highlighted by the light green box of the payoff table, and the BR strategy against it in bold, dark green.

<table>
<tr><td colspan="6" align="center">Player 2</td><td></td></tr>
</table>

|       | $A$ | $B$ | $C$ | $D$ | $X$ |
|-------|-----|-----|-----|-----|-----|
| $A$   | 0 | $-\phi$ | 1 | $\phi$ | $-\varepsilon$ |
| $B$   | $\phi$ | 0 | $-\phi^2$ | 1 | $-\varepsilon$ |
| $C$   | $-1$ | $\phi^2$ | 0 | $-\phi$ | $-\varepsilon$ |
| $D$   | $-\phi$ | $-1$ | $\phi$ | 0 | $-\varepsilon$ |
| $X$   | $\boldsymbol{\varepsilon}$ | $\boldsymbol{\varepsilon}$ | $\boldsymbol{\varepsilon}$ | $\boldsymbol{\varepsilon}$ | 0 |

Player 1 labels rows $A, B, C, D, X$.

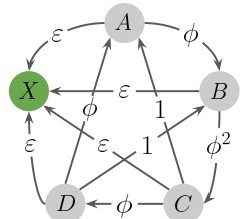

(e) The $\alpha$-Rank distribution over $\{C, D, A, B\}$ puts mass $(1/3, 1/3, 1/6, 1/6)$ on $(A, B, C, D)$ respectively. $A$ beats $C$ and $D$, and therefore its PBR score is $1/3$. $B$ beats $A$ and $D$, therefore its PBR score is $1/2$. $C$ beats $B$, its PBR score is therefore $1/3$. $D$ beats $C$, its PBR score is therefore $1/6$. Finally, $X$ beats every other strategy, and its PBR score is thus 1. There is only one strategy maximizing PBR, $X$, which is then chosen, and the SSCC of the game, recovered.

Figure A.6: Example 1 with oracle $\mathcal{O} = \text{PBR}$. Steps (a) to (d) are not shown as they are identical to their analogs in Fig. A.5.

## A.2 EXAMPLE BEHAVIOR OF PSRO(NASH, BR)

A first attempt to establish convergence to $\alpha$-Rank might involve running PSRO to convergence (until the oracle returns a strategy already in the convex hull of the known strategies), and then running $\alpha$-Rank on the resulting meta-game. However, the following provides a counterexample to this approach when using either PSRO(Nash, BR) or PSRO(Uniform, BR).

<table>
<tr><td colspan="4" align="center">Player 2</td></tr>
<tr><td></td><td>$A$</td><td>$B$</td><td>$X$</td></tr>
<tr><td>$A$</td><td>0</td><td>1</td><td>$\varepsilon$</td></tr>
<tr><td>$B$</td><td>1</td><td>0</td><td>$-\varepsilon$</td></tr>
<tr><td>$X$</td><td>$-\varepsilon$</td><td>$\varepsilon$</td><td>0</td></tr>
</table>

Player 1 (rows $A, B, X$)

(a) Example 4

<table>
<tr><td colspan="3" align="center">Player 2</td></tr>
<tr><td></td><td>$A$</td><td>$B$</td></tr>
<tr><td>$A$</td><td>-1</td><td>1</td></tr>
<tr><td>$B$</td><td>1</td><td>-1</td></tr>
<tr><td>$X$</td><td>$-\varepsilon$</td><td>$-\varepsilon/2$</td></tr>
</table>

Player 1 (rows $A, B, X$)

(b) Example 5

Table 3: Illustrative games used to analyze the behavior of PSRO in Example 4. Here, $0 < \varepsilon \ll 1$. The first game is symmetric, whilst the second is zero-sum. Both tables specify the payoff to Player 1 under each strategy profile.

**Example 4.** *Consider the two-player symmetric game specified in Table 3a. The sink strongly-connected component of the single-population response graph (and hence the $\alpha$-Rank distribution) contains all three strategies, but all NE are supported on $\{A, B\}$ only, and the best response to a strategy supported on $\{A, B\}$ is another strategy supported on $\{A, B\}$. Thus, the single-population variant of PSRO, using $\mathcal{M} \in \{\text{Nash}, \text{Uniform}\}$ with initial strategies contained in $\{A, B\}$ will terminate before discovering strategy $X$; the full $\alpha$-Rank distribution will thus* not *be recovered.*

**Example 5.** *Consider the two-player zero-sum game specified in Table 3b. All strategy profiles recieve non-zero probability in the multi-population $\alpha$-Rank distribution. However, the Nash equilibrium over the game restricted to actions $A$, $B$ for each player has a unique Nash equilibrium of $(1/2, 1/2)$. Player 1's best response to this Nash is to play some mixture of $A$ and $B$, and therefore strategy $X$ is not recovered by PSRO(Nash, BR) in this case, and so the full $\alpha$-Rank distribution will thus* not *be recovered.*

## A.3 COUNTEREXAMPLES: $\alpha$-RANK VS. NASH SUPPORT

**The Game of Chicken** The Game of Chicken provides an example where the support of $\alpha$-Rank-in the multipopulation case - does not include the full support of Nash Equilibria.

This game has three Nash equilibria: Two pure, (D,C) and (C,D), and one mixed, where the population plays Dare with probability $\frac{1}{3}$. Nevertheless, $\alpha$-rank only puts weight on (C,D) and (D,C), effectively not putting weight on the full mixed-nash support.

Player 2

|        |   | D       | C       |
|--------|---|---------|---------|
| Player 1 | D | $(0,0)$ | $(7,2)$ |
|          | C | $(2,7)$ | $(6,6)$ |

Table 4: Game of Chicken payoff table

**Prisoner's Dilemma**    The Prisoner's Dilemma provides a counterexample that the support of $\alpha$-Rank- in the multi-population case - does not include the full support of correlated equilibria.

Player 2

|          |   | D        | C        |
|----------|---|----------|----------|
| Player 1 | D | $(0,0)$  | $(3,-1)$ |
|          | C | $(-1,3)$ | $(2,2)$  |

Table 5: Prisoner's Dilemma payoff table

This game has correlated equilibria that include (C,D), (D,C) and (C,C) in their support; nevertheless, $\alpha$-Rank only puts weight on (D,D), effectively being fully disjoint from the support of the correlated equilibria.

## A.4 SINGLE-POPULATION $\alpha$-CONV

In analogy with the multi-population definition in Section 4.2, we define a single-population version of $\alpha$-CONV. We start by defining the single-population version of PBR-Score, given by PBR-SCORE$(\sigma; \boldsymbol{\pi}, S) = \sum_i \boldsymbol{\pi}_i \mathbb{1} \left[ \boldsymbol{M}^1(\sigma, s_i) > \boldsymbol{M}^2(\sigma_i, s_i) \right]$. The single-population $\alpha$-CONV is then defined as

$$\alpha\text{-CONV} = \max_{\sigma} \text{PBR-SCORE}(\sigma) - \max_{s \in S} \text{PBR-SCORE}(s),$$

where $\max_{\sigma}$ is taken over the pure strategies of the underlying game.

## B  Proofs

### B.1  Proof of Proposition 1

**Proposition 1.** *If at any point the population of $\alpha$-PSRO contains a member of an SSCC of the game, then $\alpha$-PSRO will $\alpha$-partially converge to that SSCC.*

*Proof.* Suppose that a member of one of the underlying game's SSCCs appears in the $\alpha$-PSRO population. This member will induce its own meta-SSCC in the meta-game's response graph. At least one of the members of the underlying game's corresponding SSCC will thus always have positive probability under the $\alpha$-Rank distribution for the meta-game, and the PBR oracle for this meta-SSCC will always return a member of the underlying game's SSCC. If the PBR oracle returns a member of the underlying SSCC already in the PSRO population, we claim that the corresponding meta-SSCC already contains a cycle of the underlying SSCC. To see this, note that if the meta-SSCC does not contain a cycle, it must be a singleton. Either this singleton is equal to the full SSCC of the underlying game (in which we have $\alpha$-fully converged), or it is not, in which case the PBR oracle must return a new strategy from the underlying SSCC, contradicting our assumption that it has terminated. $\square$

### B.2  Proof of Proposition 2

**Proposition 2.** *If we constrain the PBR oracle used in $\alpha$-PSRO to be novelty-bound, then $\alpha$-PSRO will $\alpha$-fully converge to at least one SSCC of the game.*

*Proof.* Suppose that $\alpha$-PSRO has converged, and consider a meta-SSCC. Since $\alpha$-PSRO has converged, it follows that each strategy profile of the meta-SSCC is an element of an SSCC of the underlying game. Any strategy profile in this SSCC which is not in the meta-SSCC will obtain a positive value for the PBR objective, and since $\alpha$-PSRO has converged, there can be no such strategy profile. Thus, the meta-SSCC contains every strategy profile contained within the corresponding SSCC of the underlying game, and therefore conclude that $\alpha$-PSRO $\alpha$-fully converges to an SSCC of the underlying game. $\square$

### B.3  Proof of Proposition 3

**Proposition 3.** *(Single-population) $\alpha$-PSRO converges $\alpha$-partially to the unique SSCC.*

*Proof.* The uniqueness of the SSCC follows from the fact that in the single-population case, the response graph is fully-connected. Suppose at termination of $\alpha$-PSRO, the $\alpha$-PSRO population contains no strategy within the SSCC, and let $s$ be a strategy in the SSCC. We claim that $s$ attains a higher value for the objective defining the PBR oracle than any strategy in the $\alpha$-PSRO population, which contradicts the fact that $\alpha$-PSRO has terminated. To complete this argument, we note that by virtue of $s$ being in the SSCC, we have $M^1(s, s') > M^1(s', s)$ for all $s'$ outside the SSCC, and in particular for all $s' \in S$, thus the PBR objective for $s$ is 1. In contrast, for any $s_i \in S$, the PBR objective for $s_i$ is upper-bounded by $1 - \pi_i$. If $\pi_i > 0$, then this shows $s_i$ is not selected by the oracle, since the objective value is lower than that of $s$. If $\pi_i = 0$, then the objective value for $s_i$ is 0, and so an SSCC member will always have a maximal PBR score of 1 against a population not composed of any SSCC member, and all members of that population have $< 1$ PBR scores. Consequently, single-population $\alpha$-PSRO cannot terminate before it has encountered an SSCC member. By Proposition 1, the proposition is therefore proven. $\square$

### B.4  Proof of Proposition 4

**Proposition 4.** *(Multi-population) Without a novelty-bound oracle, there exist games for which $\alpha$-PSRO does not converge $\alpha$-partially to any SSCC.*

*Proof.* We exhibit a specific counterexample to the claim. Consider the three-player, three-strategy game with response graph illustrated in Fig. B.7a; note that we do not enumerate all strategy profiles not appearing in the SSCC for space and clarity reasons. The sequence of updates undertaken by $\alpha$-PSRO in this game is illustrated in Figs. B.7b to B.7f; whilst the singleton strategy profile $(3, 2, 3)$

forms the unique SSCC for this game, $\alpha$-PSRO terminates before reaching it, which concludes the proof. The steps taken by the algorithm are described below; again, we do not enumerate all strategy profiles not appearing in the SSCC for space and clarity reasons.

1. Begin with strategies $[[2], [1], [1]]$ in the $\alpha$-PSRO population (Player 1 only has access to strategy 2, Players 2 and 3 only have access to strategy 1)
2. The PBR to (2,1,1) for player 2 is 2, and no other player has a PBR on this round. We add 2 to the strategy space of player 2, which changes the space of available joint strategies to $[(2, 1, 1), (2, 2, 1)]$.
3. $\alpha$-Rank puts all its mass on (2,2,1). The PBR to (2,2,1) for player 3 is 2, and no other player has a PBR on this round. We add strategy 2 to player 3's strategy space, which changes the space of available joint strategies to $[(2, 1, 1), (2, 2, 1), (2, 2, 2)]$.
4. $\alpha$-Rank puts all its mass on (2,2,2). The PBR to (2,2,2) for player 1 is 1, and no other player has a PBR on this round. We add strategy 1 to player 1's strategy space, which changes the space of available joint strategies to $[(1, 1, 1), (1, 1, 2), (1, 2, 1), (1, 2, 2), (2, 1, 1), (2, 2, 1), (2, 2, 2)]$.
5. Define $\sigma$ as the $\alpha$-Rank probabilities of the meta-game. Player 1 playing strategy 2 has a PBR score of $\sigma((1, 1, 1)) + \sigma((1, 2, 1))$, and the same player playing strategy 3 has a PBR score of $\sigma((1, 2, 1))$, which is lower than the PBR Score of playing strategy 2. No other player has a valid PBR for this round, and therefore, $\alpha$-PSRO terminates.

$\square$

In the above example, pictured in Fig. B.7, a relatively weak joint strategy (Strategy (3,2,1)) bars agents from finding the optimal joint strategy of the game (Strategy (3,2,3)) : getting to this joint strategy requires coordinated changes between agents, and is therefore closely related to the common problem of *Action/Equilibrium Shadowing* mentioned in (Matignon et al., 2012).

## B.5   PROOF OF PROPOSITION 5

**Proposition 5.** *A constant-sum game is denoted as **win-loss** if $M^k(s) \in \{0, 1\}$ for all $k \in [K]$ and $s \in S$. BR is compatible with PBR in win-loss games in the two-player single-population case.*

*Proof.* We manipulate the best-response objective as follows:

$$M^1(\nu, \pi) = \sum_{s \in S} \pi(s) M^1(\nu, s)$$
$$= \sum_{s \in S} \pi(s) \mathbb{1}[M^1(\nu, s) > M^2(\nu, s)].$$

Noting that the final line is the single-population PBR objective, we are done.  $\square$

## B.6   PROOF OF PROPOSITION 6

**Proposition 6.** *A symmetric two-player game is denoted **monotonic** if there exists a function $f : S \to \mathbb{R}$ and a non-decreasing function $\sigma : \mathbb{R} \to \mathbb{R}$ such that $M^1(s, \nu) = \sigma(f(s) - f(\nu))$. BR is compatible with PBR in monotonic games in the single-population case.*

*Proof.* Rewriting the objectives given that the game is monotonic, we have that the value-based objective becomes

$$\sum_{k=1}^{K} \pi_k M^1(s, s_k) = \sum_{k=1}^{K} \pi_k \sigma(f(s) - f(s_k)).$$

Given the fact that the only condition we have on $\sigma$ is its non-decreasing character, this objective does not reduce to maximizing $f(s)$ in the general case.

The objective for PBR is

$$\sum_{k=1}^{K} \pi_k \mathbb{1}[M^1(s, s_k) > M^2(s, s_k)] = \sum_{k=1}^{K} \pi_k \mathbb{1}[\sigma(f(s) - f(s_k)) > \sigma(f(s_k) - f(s))]$$

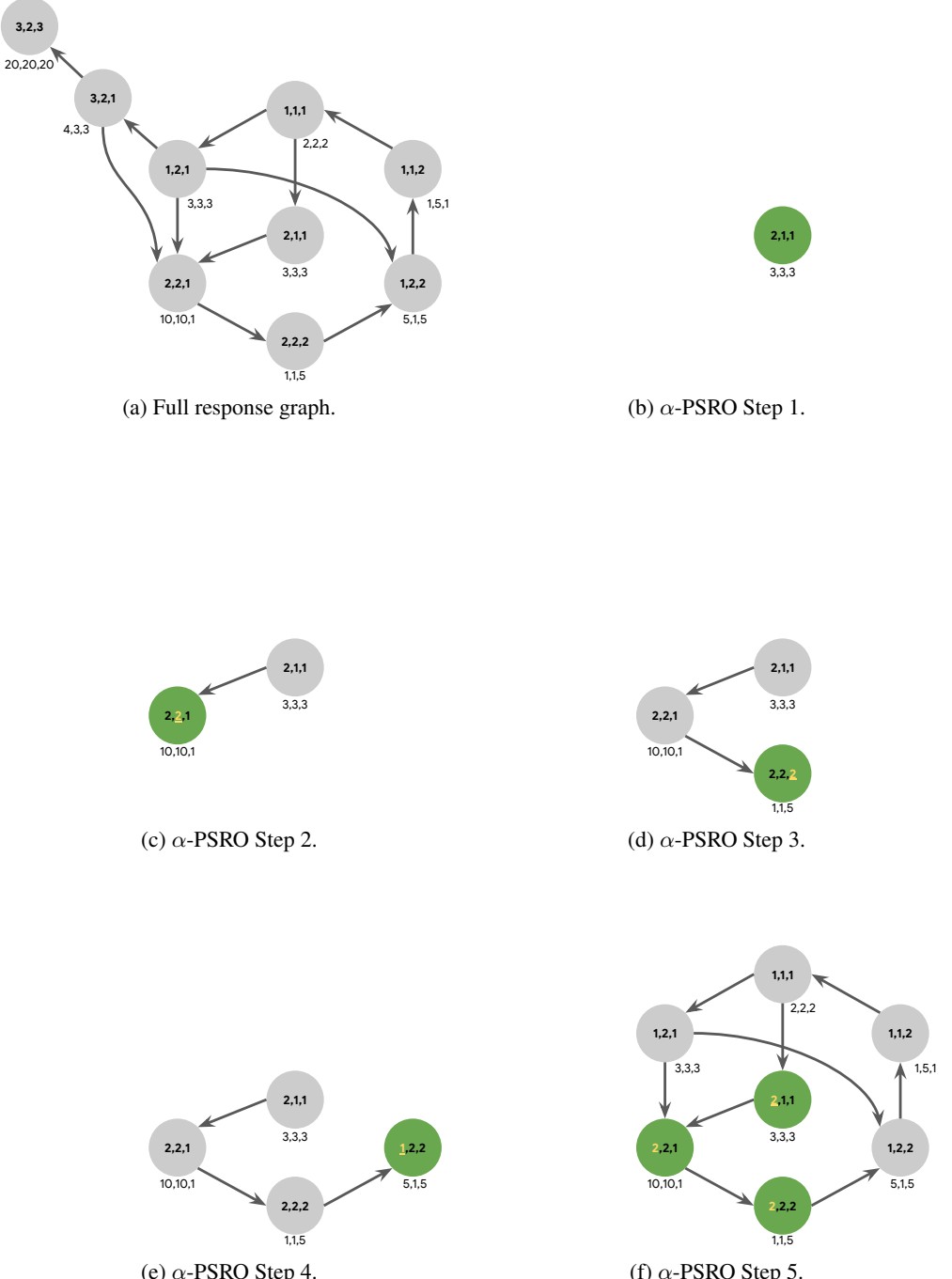

Figure B.7: The three-player, three-strategy game serving as a counterexample in the proof of Proposition 4. Strategy profiles are illustrated by gray circles, with payoffs listed beneath. All strategy profiles not pictured are assumed to be dominated, and are therefore irrelevant in determining whether $\alpha$-PSRO reaches an SSCC for this game.

Since $\sigma$ is non-decreasing,

$$\sigma(f(s) - f(s_k)) > \sigma(f(s_k) - f(s)) \;\Rightarrow\; f(s) > f(s_k)$$

and conversely,

$$f(s) > f(s_k) \;\Rightarrow\; \sigma(f(s) - f(s_k)) \geq \sigma(f(s_k) - f(s))$$

Without loss of generality, we reorder the strategies such that if $i < k$, $f(s_i) \leq f(s_k)$.

Let $s_v$ maximize the value objective. Therefore, by monotonicity, $s_v$ maximizes $\sigma(f(s) - f(s_K))$. Three possibilities then ensue.

**If there exists** $s$ such that

$$\sigma(f(s) - f(s_K)) > \sigma(f(s_K) - f(s))$$

then

$$\sigma(f(s_v) - f(s_K)) > \sigma(f(s_K) - f(s_v))$$

since $s_v$ maximizes $\sigma(f(s) - f(s_K))$ and $\sigma$ is non-decreasing. Consequently $s_v$ maximizes the PBR objective. Indeed, let us remark that for all $k \leq K$, we have that

$$\sigma(f(s_v) - f(s_k)) > \sigma(f(s_k) - f(s_v))$$

since

$$\sigma(f(s_v) - f(s_k)) \geq \sigma(f(s_v) - f(s_K)) > \sigma(f(s_K) - f(s_v)) \geq \sigma(f(s_k) - f(s_v))$$

Else, if there does not exist any policy $s$ such that $\sigma(f(s) - f(s_K)) > \sigma(f(s_K) - f(s))$, that is, for all $s$,

$$\sigma(f(s) - f(s_K)) \leq \sigma(f(s_K) - f(s))$$

Since $s_K$ is a possible solution to the value objective,

$$\sigma(f(s_v) - f(s_K)) = \sigma(f(s_K) - f(s_v))$$

Let $n$ be the integer such that

$$s_n = \arg\max\{f(s_k), s_k \in \text{ Population } \mid \exists s \text{ s.t. } \sigma(f(s) - f(s_k)) > \sigma(f(s_k) - f(s))\}$$

**If $s_n$ exists**, then we have that for all $s_i$ such that $f(s_i) > f(s_n)$,

$$\sigma(f(s_v) - f(s_i)) = \sigma(f(s_i) - f(s_v))$$

The PBR objective is

$$\sum_{k=1}^{K} \boldsymbol{\pi}_k \mathbb{1}[\sigma(f(s) - f(s_k)) > \sigma(f(s_k) - f(s))]$$

which, according to our assumptions, is equivalent to

$$\sum_{k=1}^{n} \boldsymbol{\pi}_k \mathbb{1}[\sigma(f(s) - f(s_k)) > \sigma(f(s_k) - f(s))]$$

We know that for all $i \leq n$, $\sigma(f(s_v) - f(s_i)) > \sigma(f(s_i) - f(s_v))$, and therefore, $s_v$ maximizes the PBR objective.

**Finally, if** $s_n$ doesn't exist, then any policy is solution to the PBR objective, and therefore $s_v$ is. $\quad\square$

A toy example showing the compatibility between Best Response and Preference-based Best Response is shown in Fig. B.8. The setting is that of a monotonic game where every strategy is assigned a number. Strategies are then dominated by all strategies with higher number than theirs. We compute BR and PBR on an initial population composed of one strategy that we choose to be dominated by every other strategy. Any strategy dominating the current population is a valid solution for PBR, as represented in Fig. B.8c; whereas, if we consider that the game is monotonic with $\sigma$ a strictly increasing function, only one strategy maximizes Best Response, strategy N – and it is thus the only solution of BR, as shown in Fig. B.8d.

As we can see, the solution of BR is part of the possible solutions of PBR, demonstrating the result of Proposition 6: BR is compatible with PBR in monotonic games.

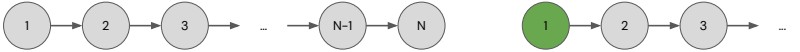

(a) Full response graph of a monotonic game.          (b) Starting population (green).

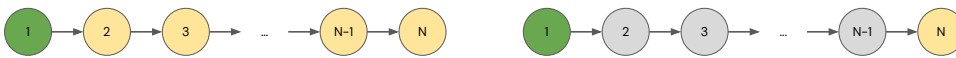

(c) Current population (green) and possible solutions   (d) Current population (green) and possible solution
of PBR (yellow)                                          of BR (yellow).

Figure B.8: Toy example of compatibility between PBR and BR: The solution returned by BR is one of the possible solutions of PBR.

## B.7 PROOF OF PROPOSITION 7

**Proposition 7.** *Consider symmetric win-loss games where outcomes between deterministic strategies are deterministic. A preference-based RL agent (i.e., an agent aiming to maximize its probability of winning against a distribution $\pi$ of strategies $\{s_1, \ldots, s_N\}$) optimizes exactly the PBR objective* (1).

*Proof.* Commencing with the above preference-based RL objective, we calculate as follows,

$$\arg\max_{\sigma} \mathbb{P}\left(\sigma \text{ beats } \sum_{i=1}^{N} \pi_i s_i\right) = \arg\max_{\sigma} \mathbb{E}_i\left[\mathbb{P}(\sigma \text{ beats } s_i | \text{index } i \text{ selected})\right]$$

$$= \arg\max_{\sigma} \sum_{i=1}^{N} \pi_i \mathbb{P}(\sigma \text{ beats } s_i)$$

$$= \arg\max_{\sigma} \sum_{i=1}^{N} \pi_i \mathbb{1}[\sigma \text{ receives a positive expected payoff against } s_i]$$

with the final equality whenever game outcomes between two deterministic strategies are deterministic. Note that this is precisely the PBR objective (1). $\qquad\square$

## B.8 PROOF OF PROPOSITION 8

**Proposition 8.** *For symmetric two-player zero-sum games where off-diagonal payoffs have equal magnitude, all NE have support contained within that of the single-population $\alpha$-Rank distribution.*

*Proof.* In the single-population case, the support of the $\alpha$-Rank distribution is simply the (unique) sink strongly-connected component of the response graph (uniqueness follows from the fact that the response graph, viewed as an undirected graph, is fully-connected). We will now argue that for a strategy $s$ in the sink strongly-connected component and a strategy $z$ outside the sink strongly-connected component, we have

$$\sum_{a \in S} \pi(a) M^1(s, a) > \sum_{a \in S} \pi(a) M^1(z, a), \tag{3}$$

This inequality states that when an opponent plays according to $\pi$, the expected payoff to the row player is greater if they defect to $s$ whenever they would have played $z$. This implies that if a supposed symmetric Nash equilibrium contains a strategy $z$ outside the sink strongly-connected component in its support, then it could receive higher reward by playing $s$ instead, which contradicts the fact that it is an NE. We show (3) by proving a stronger result — namely, that $s$ dominates $z$ as strategies. Firstly, since $s$ is the sink strongly-connected component and $z$ is not, $s$ beats $z$, and so $M^1(s, z) > M^1(s, s) = M^1(z, z) > M^1(z, s)$. Next, if $a \notin \{s, z\}$ is in the sink strongly-connected component, then $a$ beats $z$, and so $M^1(s, a) > M^1(z, a)$ if $s$ beats $a$, and $M^1(s, a) = M^1(z, a)$ otherwise. Finally, if $a \neq s, z$ is not in the sink strongly-connected component, then $M^1(s, a) = M^1(z, a)$ is $z$ beats $a$, and $M^1(s, a) > M^1(z, a)$ otherwise. Thus, (3) is proven, and the result follows. $\qquad\square$

## B.9 PROOF OF PROPOSITION 9

**Proposition 9.** *In a symmetric two-player zero-sum game, there exists an NE with support contained within that of the $\alpha$-Rank distribution.*

*Proof.* Consider the restriction of the game to the strategies contained in the sink strongly-connected component of the original game. Let $\pi$ be an NE for this restricted game, and consider this as a distribution over all strategies in the original game (putting $0$ mass on strategies outside the sink component). We argue that this is an NE for the full game, and the statement follows. To see this, note that since any strategy outside the sink strongly-connected component receives a non-positive payoff when playing against a strategy in the sink strongly-connected component, and that for at least one strategy in the sink strongly-connected component, this payoff is negative. Considering the payoffs available to the row player when the column player plays according to $\pi$, we observe that the expected payoff for any strategy outside the sink strongly-connected component is negative, since every strategy in the sink strongly-connected component beats the strategy outside the component. The payoff when defecting to a strategy in the sink strongly-connected component must be non-positive, since $\pi$ is an NE for the restricted game. $\square$

# C  ADDITIONAL DETAILS ON EXPERIMENTS

## C.1  EXPERIMENTAL PROCEDURES

The code backend for the Poker experiments used OpenSpiel (Lanctot et al., 2019). Specifically, we used OpenSpiel's Kuhn and Leduc poker implementations, and exact best responses were computed by traversing the game tree (see implementation details in `https://github.com/deepmind/open_spiel/blob/master/open_spiel/python/algorithms/best_response.py`). 100 game simulations were used to estimate the payoff matrix for each possible strategy pair.

Although the underlying Kuhn and Leduc poker games are stochastic (due to random initial card deals), the associated meta-games are essentially deterministic (as, given enough game simulations, the mean payoffs are fixed). The subsequent PSRO updates are, thus, also deterministic.

Despite this, we report averages over 2 runs per PSRO $\mathcal{M}$, primarily to capture stochasticity due to differences in machine-specific rounding errors that occur due to the distributed computational platforms we run these experiments on.

For experiments involving $\alpha$-Rank, we conduct a full sweep over the ranking-intensity parameter, $\alpha$, following each iteration of $\alpha$-PSRO. We implemented a version of $\alpha$-Rank (building on the Open-Spiel implementation `https://github.com/deepmind/open_spiel/blob/master/open_spiel/python/egt/alpharank.py`) that used a sparse representation for the underlying transition matrix, enabling scaling-up to the large-scale NFG results presented in the experiments.

For experiments involving the projected replicator dynamics (PRD), we used uniformly-initialized meta-distributions, running PRD for $5e4$ iterations, using a step-size of `dt` $= 1e - 3$, and exploration parameter $\gamma = 1e - 10$. Time-averaged distributions were computed over the entire trajectory.

## C.2  DOMAIN DESCRIPTION AND GENERATION

### C.2.1  NORMAL FORM GAMES GENERATION

Algorithms 2 to 4 provide an overview of the procedure we use to randomly-generate normal-form games for the oracle comparisons visualized in Fig. 2.

---

**Algorithm 2** GenerateTransitive(Actions, Players, mean$_\text{value}$ = $[0.0, 1.0]$, mean$_\text{probability}$ = $[0.5, 0.5]$, var $= 0.1$)

---

1: $\mathcal{T} = []$
2: **for** Player $k$ **do**
3:     Initialize $f_k = [0] * $ Actions
4:     **for** Action $a \leq$ Actions **do**
5:         Randomly sample mean $\mu$ from mean$_\text{value}$ according to mean$_\text{probability}$
6:         $f_k[a] \sim \mathcal{N}(\mu, \text{var})$
7: **for** Player $k$ **do**
8:     $\mathcal{T}[k] = f_k - \frac{1}{|\text{Players}|-1} \sum_{i \neq k} f_i$
9: Return $\mathcal{T}$

---

**Algorithm 3** GenerateCyclic(Actions, Players, var $= 0.4$)

---

1: $\mathcal{C} = []$
2: **for** Player $k$ **do**
3:     Initialize $C[k] \sim \mathcal{N}(0, \text{var})$, Shape$(C[k]) = (\text{Actions}_\text{First Player}, \ldots, \text{Actions}_\text{Last Player})$
4: **for** Player $k$ **do**
5:     Sum $= \sum_{\text{Actions } a_i \text{ of all player } i \neq k} C[k][a_1, \ldots, a_{k-1}, :, a_{k+1}, \ldots]$
6:     Shape(Sum) $= (1, \ldots, 1, \text{Actions}_{\text{Player } k}, 1, \ldots, 1)$
7:     $C[k] = C[k] - $ Sum
8: Return $\mathcal{C}$

---

---

**Algorithm 4** General Normal Form Games Generation(Actions, Players)

---

1: Generate matrix lists $\mathcal{T}$ = GenerateTransitive(Actions, Players), $\mathcal{C}$ = GenerateCyclic(Actions, Players)
2: Return $[\mathcal{T}[k] + \mathcal{C}[k]$ for Player k$]$

---

### C.2.2 KUHN AND LEDUC POKER

$K$-player Kuhn poker is played with a deck of $K + 1$ cards. Each player starts with 2 chips and 1 face-down card, and antes 1 chip to play. Players either bet (raise/call) or fold iteratively, until each player is either in (has contributed equally to the pot) or has folded. Amongst the remaining players, the one with the highest-ranked card wins the pot.

Leduc Poker, in comparison, has a significantly larger state-space. Players in Leduc have unlimited chips, receive 1 face-down card, ante 1 chip to play, with subsequent bets limited to 2 and 4 chips in rounds 1 and 2. A maximum of two raises are allowed in each round, and a public card is revealed before the second round.

### C.3 PBR COMPUTATION IN NORMAL FORM GAMES

The algorithms used to compute PBR and PBR-SCORE in the games generated by the algorithm described in Section C.2.1 is shown in Algorithms 5 and 6. Note that they compute the multipopulation version of PBR. PCS-SCORE is computed by pre-computing the full game's SSCC, and computing the proportion of currently selected strategies in the empirical game that also belongs to the full game's SSCC.

Note that the PBR-SCORE and PCS-SCORE are useful measures for assessing the quality of convergence in our examples, in a manner analogous to NASHCONV. The computation of these scores is, however, not tractable in general games. Notably, this is also the case for NASHCONV (as it requires computation of player-wise best responses, which can be problematic even in moderately-sized games). Despite this, these scores remain a useful way to empirically verify the convergence characteristics in small games where they can be tractably computed.

---

**Algorithm 5** PBR Score(Strategy S, Payoff Tensor, Current Player Id, Joint Strategies, Joint Strategy Probability)

---

1: New strategy score = 0
2: **for** Joint strategy J, Joint probability P in Joint Strategies, Joint Strategy Probability **do**
3:     New strategy = J
4:     New strategy[Current Player Id] = S
5:     New strategy payoff = Payoff Tensor[New Strategy]
6:     Old strategy payoff = Payoff Tensor[J]
7:     New strategy score += P * (New Strategy Payoff > Old Strategy Payoff)
8: Return New strategy score

---

**Algorithm 6** PBR(Payoff Tensor list LM, Joint Strategies per player PJ, Alpharank Probability per Joint Strategy PA, Current Player)

---

1: $\max_{PBR} = 0$
2: $\max_{strat} = None$
3: **for** Strategy S available to Current Player among all possible strategies **do**
4:     score = PBR Score(S, LM[Current Player Id], Current Player Id, PJ, PA)
5:     **if** New Strategy Score $> \max_{PBR}$ **then**
6:         $\max_{PBR}$ = New Strategy Score
7:         $\max_{strat}$ = S
8: Return $\max_{PBR}, \max_{strat}$

---

## C.4 Additional Oracle Comparison Results

We present additional oracle comparisons in Fig. C.9, all of these in the multi-population case.

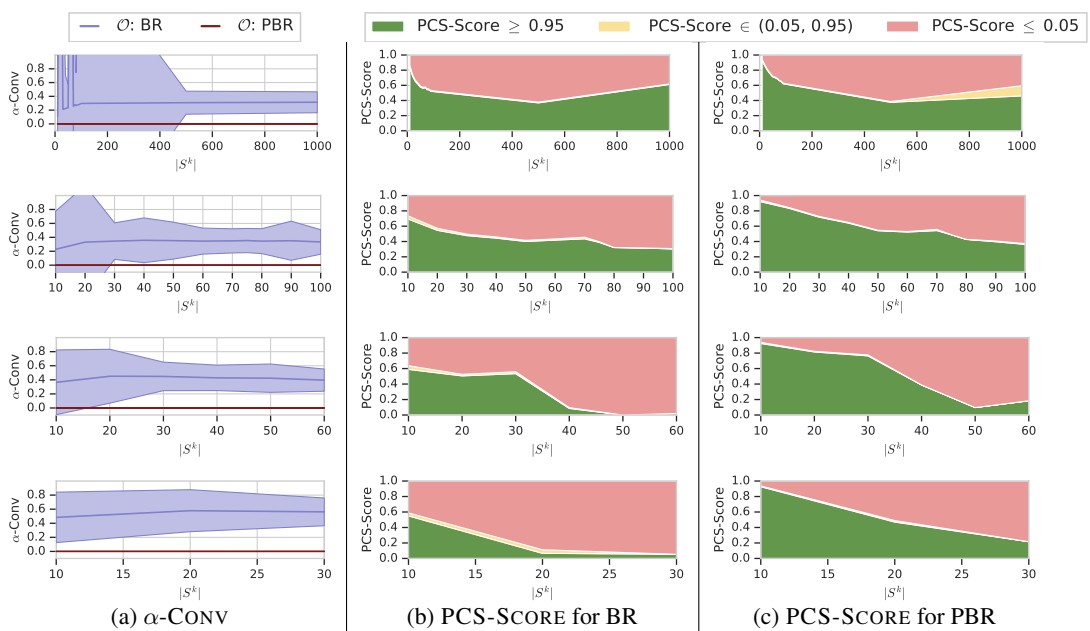

(a) $\alpha$-Conv        (b) PCS-Score for BR        (c) PCS-Score for PBR

Figure C.9: Oracle comparisons for randomly-generated normal-form games with varying player strategy space sizes $|S^k|$. The rows, in order, correspond to 2- to 5-player games.

## C.5 Notes on Rectified Nash performance

This section provides additional insights into the Rectified Nash results detailed in Section 5. We begin with an important disclaimer that Rectified Nash was developed solely with symmetric games in mind. As Kuhn Poker and Leduc Poker are not symmetric games, they lie beyond the theoretical scope of Rectified Nash. Nevertheless, comparing the performance of rectified and non-rectified approaches from an empirical perspective yields insights, which may be useful for future investigations that seek to potentially extend and apply rectified training approaches to more general games.

As noted in the main paper, the poor performance of PSRO using Rectified Nash (in Fig. 3) is initially surprising as it indicates premature convergence to a high-NashConv distribution over the players' policy pools. Investigating this further led to a counterintuitive result for the domains evaluated: Rectified Nash was, roughly speaking, not increasing the overall diversity of behavioral policies added to each player's population pool. In certain regards, it even prevented diversity from emerging.

To more concretely pinpoint the issues, we detail below the first 3 iterations of PSRO(Rectified Nash, BR) in Kuhn Poker. Payoff matrices at each PSRO iteration are included in Tables 6a to 6c. For clarity, we also include the 5 best responses trained by Rectified Nash and the policies they were trained against, in their order of discovery: 2 policies for Player 1 (in Fig. C.11) and 3 policies for Player 2 (in Fig. C.12).

1. Iteration 0: both players start with uniform random policies.
2. Iteration 1:
    - Player 1 trains a best response against Player 2's uniform random policy; its policy set is now the original uniform policy, and the newly-computed best response.
    - Player 2 trains a best response against Player 1's uniform random policy; its policy set is now the original uniform policy, and the newly-computed best response.
    - Player 2's best response beats both of Player 1's policies.
    - Payoff values are represented in Table 6a.

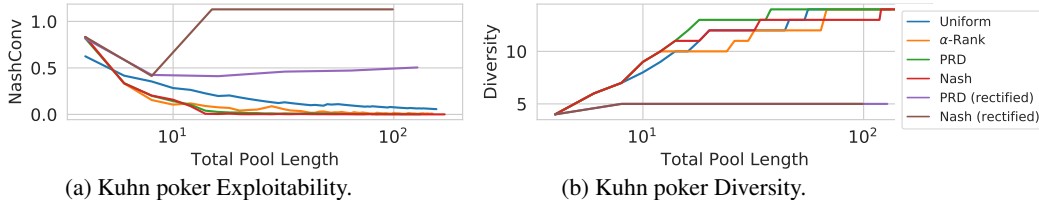

(a) Kuhn poker Exploitability.  (b) Kuhn poker Diversity.

Figure C.10: Policy Exploitability and Diversity in 2-player Kuhn for a given seed and 100 simulations per payoff entry.

3. Iteration 2:
   - By Rectified Nash rules, Player 1 only trains policies against policies it beats; i.e., only against Player 2's random policy, and thus it adds the same policy as in iteration 1 to its pool.
   - Player 2 trains a best response against the Nash mixture of Player 1's first best response and random policy. This policy also beats all policies of player 1.
   - Payoff values are represented in Table 6b.

4. Iteration 3:
   - Player 1 only trains best responses against Player 2's random policy.
   - Player 2 only trains best responses against the Nash of Player 1's two unique policies. This yields the same policies for player 2 as those previously added to its pool (i.e., a loop occurs).
   - Payoff values are represented in Table 6c

5. Rectified Nash has looped.

As noted above, Rectified Nash loops at iteration 3, producing already-existing best responses against Player 1's policies. Player 1 is, therefore, constrained to never being able to train best responses against any other policy than Player 2's random policy. In turn, this prevents Player 2 from training additional novel policies, and puts the game in a deadlocked state.

Noise in the payoff matrices may lead to different best responses against the Nash Mixture of policies, effectively increasing diversity. However, this effect did not seem to manifest in our experiments. To more clearly illustrate this, we introduce a means of evaluating the policy pool diversity, counting the number of unique policies in the pool. Specifically, given that Kuhn poker is a finite state game, comparing policies is straightforward, and only amounts to comparing each policy's output on all states of the games. If two policies have exactly the same output on all the game's states, they are equal; otherwise, they are distinct. We plot in Fig. C.10 the policy diversity of each meta-solver, where we observe that both Rectified Nash and Rectified PRD discover a total of 5 different policies. We have nevertheless noticed that in a few rare seeds, when using low number of simulations per payoff entry (Around 10), Rectified Nash was able to converge to low exploitability scores, suggesting a relationship between payoff noise, uncertainty and convergence of Rectified Nash whose investigation we leave for future work. We also leave the investigation of the relationship between Policy Diversity and Exploitability for future work, though note that there appears to be a clear correlation between both. Overall, these results demonstrate that the Rectified Nash solver fails to discover as many unique policies as the other solvers, thereby plateauing at a low NASHCONV.

Finally, regarding Rectified PRD, which performs better in terms of NASHCONV when compared to Rectified Nash, we suspect that payoff noise *in combination* with the intrinsic noise of PRD, plays a key factor - but those two are not enough to deterministically make Rectified PRD converge to 0 exploitability, since in the seed that generated Fig. C.10, it actually doesn't (Though it indeed converges in Fig. 3). We conjecture this noisier behavior may enable Rectified PRD to free itself from deadlocks more easily, and thus discover more policies on average. A more detailed analysis of Rectified PRD is left as future work.

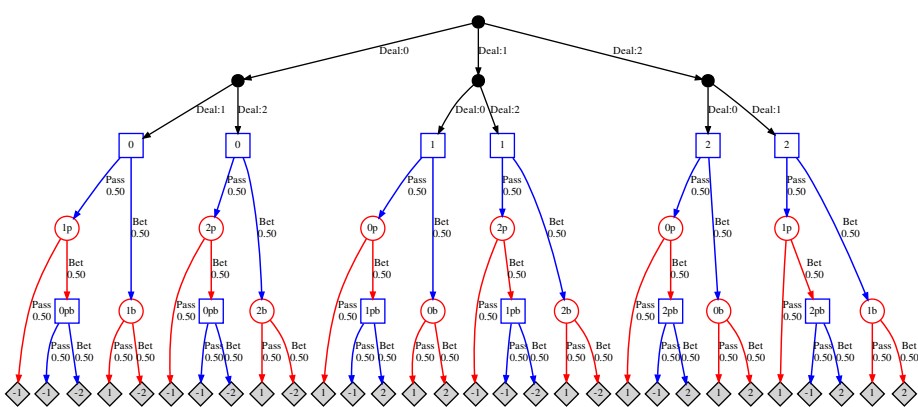

(a) Initial (uniform) policies.

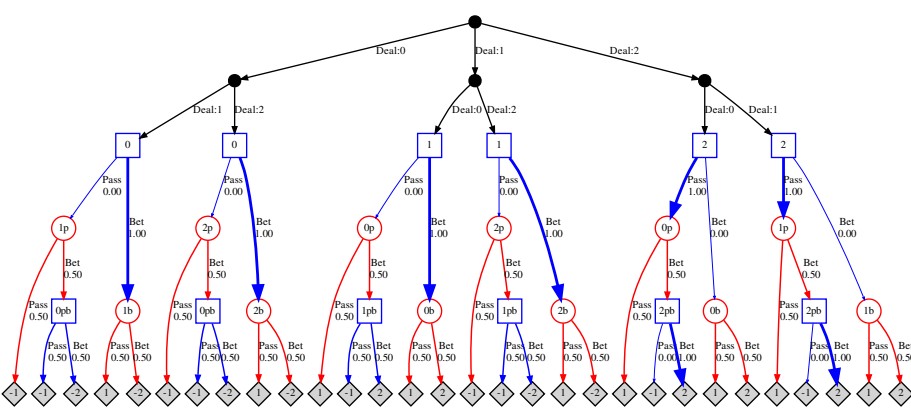

(b) Player 1's first best response indicated in blue, and the policy it best-responded against in red.

Figure C.11: Game tree with both players' policies visualized. Player 1 decision nodes and action probabilities indicated, respectively, by the blue square nodes and blue arrows. Player 2's are likewise shown via the red counterparts.

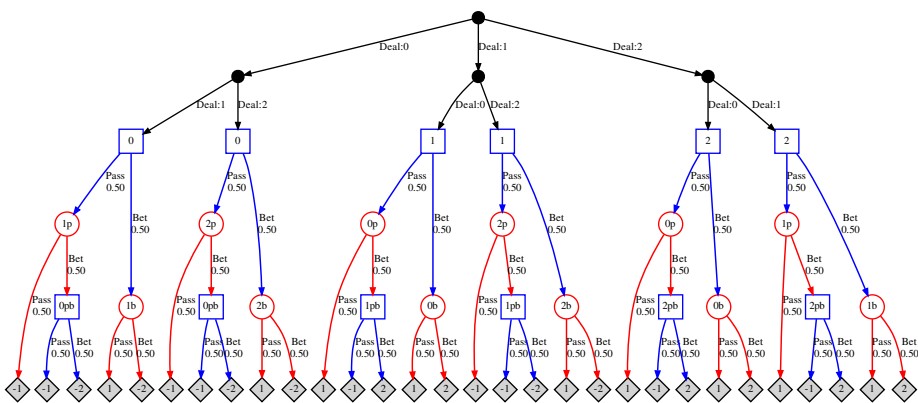

(a) Initial (uniform) policies.

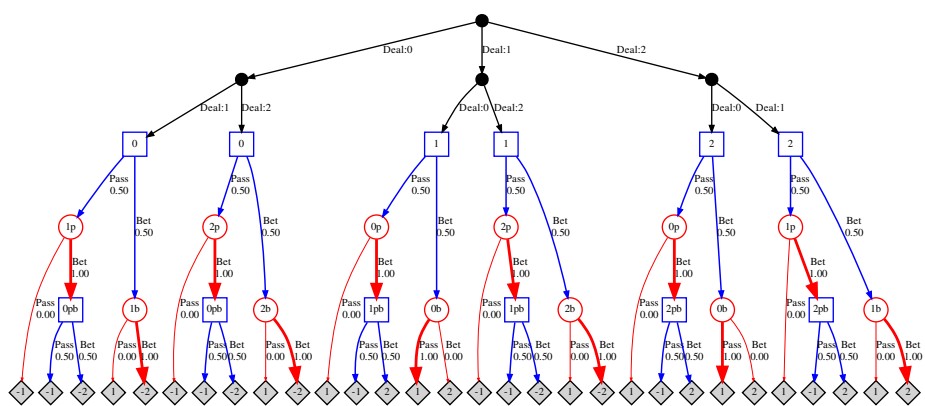

(b) Player 2's first best response indicated in red, and the policy it best-responded against in blue.

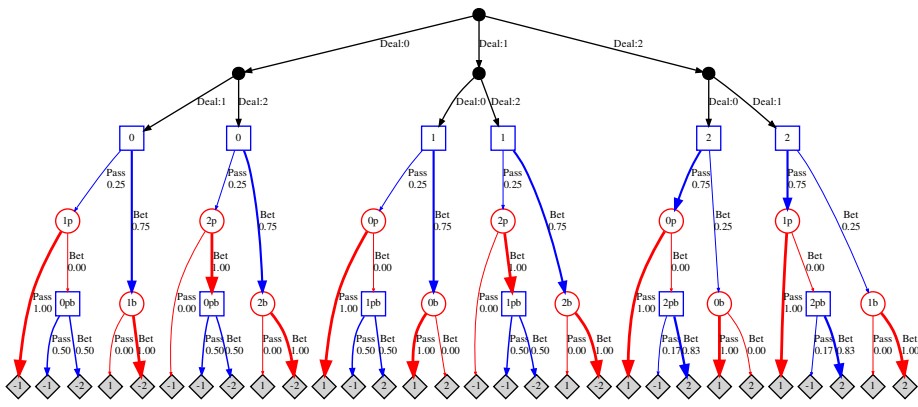

(c) Player 2's second best response indicated in red, and the policy it best-responded against in blue.

Figure C.12: Game tree with both players' policies visualized. Player 1 decision nodes and action probabilities indicated, respectively, by the blue square nodes and blue arrows. Player 2's are likewise shown via the red counterparts.

$$\begin{bmatrix} 0.1014 & -0.4287 \\ 0.4903 & -0.1794 \end{bmatrix}$$

(a) Iteration 1.

$$\begin{bmatrix} 0.1014 & -0.4287 & -0.2461 & -0.2284 \\ 0.4903 & -0.1794 & -0.4988 & -0.5228 \\ 0.5169 & -0.1726 & -0.4946 & -0.5 \\ 0.5024 & -0.1832 & -0.4901 & -0.5066 \end{bmatrix}$$

(b) Iteration 2.

$$\begin{bmatrix} 0.1014 & -0.4287 & -0.2461 & -0.2284 & -0.264 & -0.2602 & -0.2505 \\ 0.4903 & -0.1794 & -0.4988 & -0.5228 & -0.5015 & -0.5501 & -0.5159 \\ 0.5169 & -0.1726 & -0.4946 & -0.5 & -0.5261 & -0.5279 & -0.4979 \\ 0.5024 & -0.1832 & -0.4901 & -0.5066 & -0.5069 & -0.4901 & -0.5033 \\ 0.4893 & -0.1968 & -0.5084 & -0.4901 & -0.5015 & -0.4883 & -0.4796 \\ 0.4841 & -0.1496 & -0.4892 & -0.491 & -0.4724 & -0.4781 & -0.5087 \\ 0.5179 & -0.1769 & -0.503 & -0.521 & -0.4991 & -0.4739 & -0.4649 \\ 0.4959 & -0.1613 & -0.5123 & -0.518 & -0.5126 & -0.5039 & -0.4853 \end{bmatrix}$$

(c) Iteration 3.

Table 6: PSRO(Rectified Nash, BR) evaluated on 2-player Kuhn Poker. Player 1's payoff matrix shown for each respective training iteration.

# D   $\alpha$-RANK IN DETAIL

In this section we give further details of $\alpha$-Rank; for a full description, see Omidshafiei et al. (2019). Essentially $\alpha$-Rank defines a directed *response graph* over the pure strategy profiles of the game under study, by indicating when a player has an incentive to make a unilateral deviation from their current strategy. An irreducible (noisy) random walk over this graph is then defined, and the strategy profile rankings are obtained by ordering the masses of this Markov chain's unique invariant distribution $\boldsymbol{\pi}$.

The Markov transition matrix $\mathbf{C}$ that specifies this random walk is defined as follows for the multi-population case; see Omidshafiei et al. (2019) for the single-population case. Consider a pure strategy profile $s \in S$, and let $\sigma = (\sigma^k, s^{-k})$ be the pure strategy profile which is equal to $s$, except for player $k$, which uses strategy $\sigma^k \in S^k$ instead of $s^k$. Let $\mathbf{C}_{s,\sigma}$ denote the transition probability from $s$ to $\sigma$, and $\mathbf{C}_{s,s}$ the self-transition probability of $s$, with each defined as:

$$\mathbf{C}_{s,\sigma} = \begin{cases} \eta \frac{1-\exp\left(-\alpha\left(\mathbf{M}^k(\sigma)-\mathbf{M}^k(s)\right)\right)}{1-\exp(-\alpha m(\mathbf{M}^k(\sigma)-\mathbf{M}^k(s)))} & \text{if } \mathbf{M}^k(\sigma) \neq \mathbf{M}^k(s) \\ \frac{\eta}{m} & \text{otherwise ,} \end{cases}$$

$$\mathbf{C}_{s,s} = 1 - \sum_{\substack{k\in[K] \\ \sigma|\sigma^k\in S^k\backslash\{s^k\}}} \mathbf{C}_{s,\sigma} ,$$

where $\eta = (\sum_l(|S^l| - 1))^{-1}$. If two strategy profiles $s$ and $s'$ differ in more than one player's strategy, then $\mathbf{C}_{s,s'} = 0$. Here $\alpha \geq 0$ and $m \in \mathbb{N}$ are parameters to be specified; the form of this transition probability is described by evolutionary dynamics models from evolutionary game theory and is explained in more detail in Omidshafiei et al. (2019). Large values of $\alpha$ correspond to higher *selection pressure* in the evolutionary model under consideration; the version of $\alpha$-Rank used throughout this paper corresponds to the limiting invariant distribution as $\alpha \to \infty$, under which only strategy profiles appearing in the sink strongly-connected components of the response graph can have positive mass.

# E  TOWARDS THEORETICAL GUARANTEES FOR THE PROJECTED REPLICATOR DYNAMICS

Computing Nash equilibria is intractable for general games and can suffer from a selection problem (Daskalakis et al., 2009); therefore, it quickly becomes computationally intractable to employ an exact Nash meta-solver in the inner loop of a PSRO algorithm. To get around this, Lanctot et al. (2017) use regret minimization algorithms to attain an approximate correlated equilibrium (which is guaranteed to be an approximate Nash equilibrium under certain conditions on the underlying game, such as two-player zero-sum). A dynamical system from evolutionary game theory that also converges to equilibria under certain conditions is the *replicator dynamics* (Taylor and Jonker, 1978; Schuster and Sigmund, 1983; Cressman and Tao, 2014; Bloembergen et al., 2015), which defines a dynamical system over distributions of strategies $(\pi_s^k(t) \mid k \in [K], s \in S^k)$, given by

$$\dot{\pi}_s^k(t) = \pi_s^k(t) \left[ M^k(s, \pi^{-k}(t)) - M^k(\pi^k(t)) \right], \quad \text{for all } k \in [K], \ s \in S^k, \tag{4}$$

with an arbitrary initial condition. Lanctot et al. (2017) introduced a variant of replicator dynamics, termed *projected replicator dynamics* (PRD), which projects the flow of the system so that each distribution $\pi^k(t)$ lies in the set $\Delta_{S^k}^{\gamma} = \{\pi \in \Delta_{S^k} \mid \pi_s \geq \gamma/(|S^k| + 1), \forall s \in S^k\}$; see, e.g., Nagurney and Zhang (2012) for properties of such projected dynamical systems. This heuristically enforces additional "exploration" relative to standard replicator dynamics, and was observed to provide strong empirical results when used as a meta-solver within PSRO. However, the introduction of projection potentially severs the connection between replicator dynamics and Nash equilibria, and the theoretical game-theoretic properties of PRD were left open in Lanctot et al. (2017).

Here, we take a first step towards investigating theoretical guarantees for PRD. Specifically, we highlight a possible connection between $\alpha$-Rank, the calculation of which requires no simulation, and a constrained variant of PRD, which we denote the 'single-mutation PRD' (or s-PRD), leaving formal investigation of this connection for future work.

Specifically, s-PRD is a dynamical system over distributions $(\pi_s^k(t) \mid k \in [K], s \in S^k)$ that follows the replicator dynamics (equation 4), with initial condition restricted so that each $\pi_0^k$ lies on the 1-skeleton $\Delta_{S^k}^{(1)} = \{\pi \in \Delta_{S^k} \mid \sum_{s \in S^k} \mathbb{1}_{\pi_s \neq 0} \leq 2\}$. Further, whenever a strategy distribution $\pi_t^k$ enters a $\delta$-corner of the simplex, defined by $\Delta_{S^k}^{[\delta]} = \{\pi \in \Delta_{S^k}^{(1)} \mid \exists s \in S^k \text{ s.t. } \pi_s \geq 1 - \delta\}$, the non-zero element of $\pi^k(t)$ with mass at most $\delta$ is replaced with a uniformly randomly chosen strategy after a random time distributed according to $\text{Exp}(\mu)$, for some small $\mu > 0$. This concludes the description of s-PRD. We note at this stage that s-PRD defines, essentially, a dynamical system on the 1-skeleton (or edges) of the simplex, with random mutations towards a uniformly-sampled randomly strategy profile $s$ at the simplex vertices. At a high-level, this bears a close resemblance to the finite-population $\alpha$-Rank dynamics defined in Omidshafiei et al. (2019); moreover, we note that the connection between s-PRD and true $\alpha$-Rank dynamics becomes even more evident when taking into account the correspondence between the standard replicator dynamics and $\alpha$-Rank that is noted in Omidshafiei et al. (2019, Theorem 2.1.4).

We conclude by noting a major limitation of both s-PRD and PRD, which can limit their practical applicability even assuming a game-theoretic grounding can be proven for either. Specifically, with all such solvers, simulation of a dynamical system is required to obtain an approximate equilibrium, which may be costly in itself. Moreover, their dynamics can be chaotic even for simple instances of two-player two-strategy games (Palaiopanos et al., 2017). In practice, the combination of these two limitations may completely shatter the convergence properties of these algorithms in practice, in the sense that the question of *how long to wait until convergence* becomes increasingly difficult (and computationally expensive) to answer. By contrast, $\alpha$-Rank does not rely on such simulations, thereby avoiding these empirical issues.

We conclude by remarking again that, albeit informal, these results indicate a much stronger theoretical connection between $\alpha$-Rank and standard PRD that may warrant future investigation.

## F    MuJoCo Soccer Experiment

While the key objective of this paper is to take a first step in establishing a theoretically-grounded framework for PSRO-based training of agents in many-player settings, an exciting question concerns the behaviors of the proposed $\alpha$-Rank-based PSRO algorithm in complex domains where function-approximation-based policies need to be relied upon for generalizable task execution. In this section, we take a preliminary step towards conducting this investigation, focusing in particular on the MuJoCo soccer domain introduced in Liu et al. (2019) (refer to `https://github.com/deepmind/dm_control/tree/master/dm_control/locomotion/soccer` for the corresponding domain code).

In particular, we conduct two sets of initial experiments. The first set of experiments compares the performance of PSRO($\alpha$-Rank, RL) and PSRO(Uniform, RL) in games of 3 vs. 3 MuJoCo soccer, and the second set compares PSRO($\alpha$-Rank, RL) against a population-based training pipeline in 2 vs. 2 games.

### F.1    Training Procedure

For each of the PSRO variants considered, we compose a hierarchical training procedure composed of two levels. At the low-level, which focuses on simulations of the underlying MuJoCo soccer game itself, we consider a collection of 32 reinforcement learners (which we call agents) that are all trained at the same time, as in Liu et al. (2019). We compose teams corresponding to multiple clones of agent per team (yielding homogeneous teams, in contrast to (Liu et al., 2019), which evaluates teams of heterogeneous agents) and evaluate all pairwise team match-ups. Note that this yields a 2-"player" meta-game (where each "player" is actually a team, i.e., a team-vs.-team setting), with payoffs corresponding to the average win-rates of each team when pitted against each other.

The payoff matrix is estimated by simulating matches between different teams. The number of simulations per entry is adaptive based on the empirical uncertainty observed on the pair-wise match outcomes. In practice, we observed an average of 10 to 100 simulations per entry, with fewer simulations used for meta-payoffs with higher certainty. For the final evaluation matrix reported in Appendix F (Fig. F.10), which was computed after the conclusion of PSRO-based training, 100 simulations were used per entry. Additionally, instead of adding one policy per PSRO iteration per player we add three (which corresponds to the 10% best RL agents).

Several additional modifications were made to standard PSRO to help with the inherently more difficult nature of Deep Reinforcement Learning training:

- Agent performance, used to choose which agents out of the 32 to add to the pool, is measured by the $\alpha$-Rank-average for PSRO($\alpha$-Rank, RL) and Nash-average for PSRO(Uniform, RL) of agents in the (Agents, Pool) vs (Agents, Pool) game.

- Each oracle step in PSRO is composed of 1 billion learning steps of the agents. After each step, the top 10% of agents (the 3 best agents) are added to the pool, and training of the 32 agents continues;

- We use a 50% probability of training using self-play (the other 50% training against the distribution of the pool of agents).

### F.2    Results

In the first set of experiments, we train the PSRO($\alpha$-Rank, RL) and PSRO(Uniform, RL) agents independently (i.e., the two populations never interact with one another). Following training, we compare the effective performance of these two PSRO variants by pitting their 8 best trained agents against one another, and recording the average win rates. These results are reported in Fig. F.13 for games involving teams of 3 vs. 3. It is evident from these results that PSRO($\alpha$-Rank, RL) significantly outperforms PSRO(Uniform, RL). This is clear from the colorbar on the far right of Fig. F.13, which visualizes the post-training alpharank distribution over the payoff matrix of the metagame composed of both training pipelines.

In the second set of experiments, we compare $\alpha$-PSRO-based training to self-play-based training (i.e., sampling opponents uniformly at random from the training agent population). This provides a means

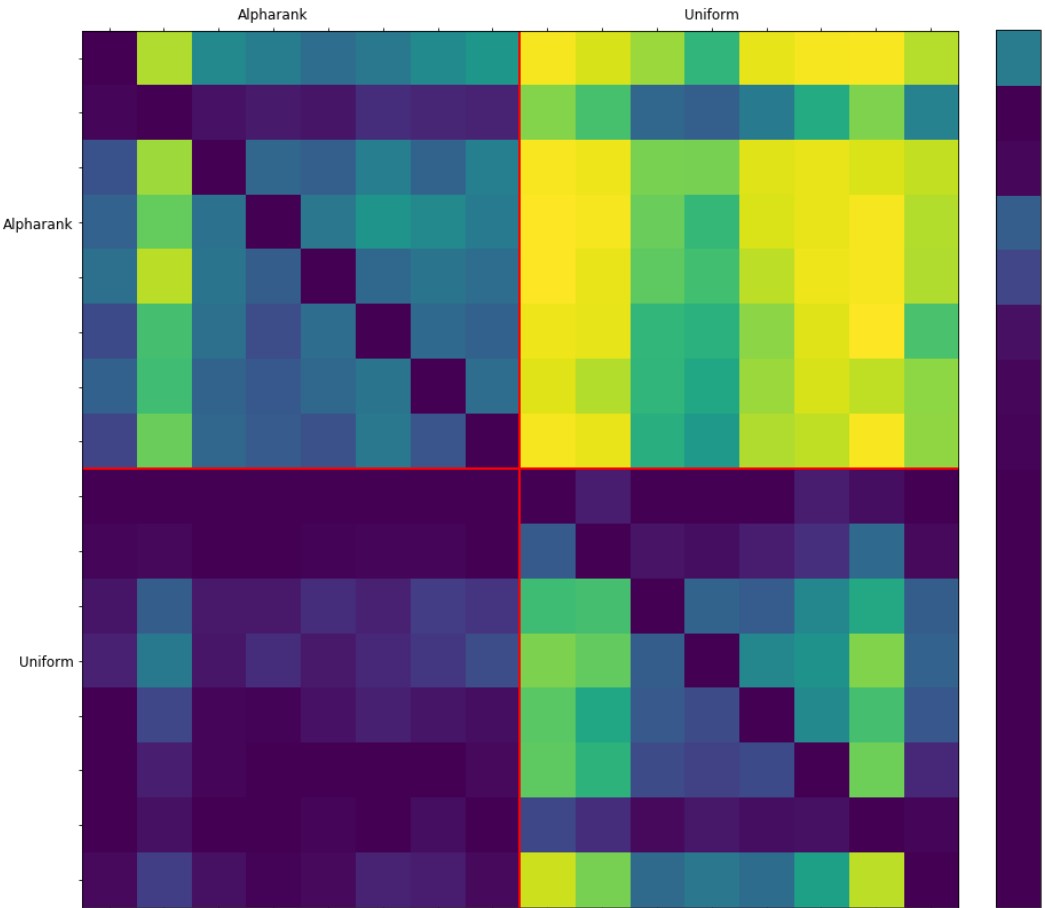

Figure F.13: $\alpha$-PSRO versus PSRO(Uniform, BR) in the MuJoCo Soccer domain. Left is the matrix representing the probability of winning for $\alpha$-PSRO and PSRO(Uniform, BR)'s best 8 agents. Right is the $\alpha$-Rank distribution over the meta-game induced by these agents. Yellow are high probabilities, dark-blue are low probabilities.The diagonal is taken to be 0.

of gauging the performance improvement solely due to PSRO; these results are reported in Fig. F.14 for games involving teams of 2 vs. 2.

We conclude by remarking that these results, although interesting, primarily are intended to lay the foundation for use of $\alpha$-Rank as a meta-solver in complex many agent domains where RL agents serve as useful oracles; additionally, more extensive research and analysis is necessary to make these results conclusive in domains such as MuJoCo soccer. We plan to carry out several experiments along these lines, including extensive ablation studies, in future work.

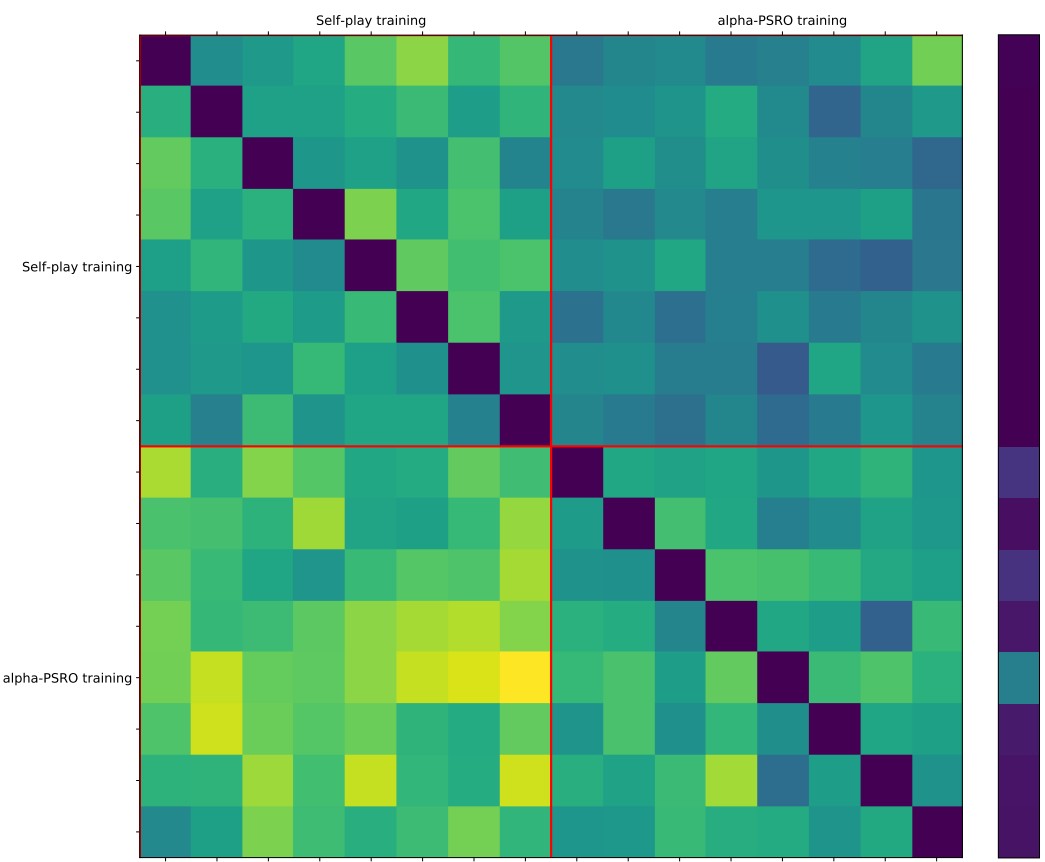

Figure F.14: $\alpha$-PSRO training pipeline vs. training pipeline without PSRO.

