# OpenReview forum: "A Generalized Training Approach for Multiagent Learning"
_ICLR.cc/2020/Conference — Accept (Talk)_

### Official Review · AnonReviewer3 · 2019-10-22
**Official Blind Review #3**

**Rating:** 8

**Review:**

The paper studies α-Rank, a scalable alternative to Nash equilibrium, across a number of areas. Specifically the paper establishes connections between Nash and α-Rank in specific instances, presents a novel construction of best response that guarantees convergence to the α-Rank in several games, and demonstrates empirical results in poker and soccer games.

The paper is well-written and well-argued. Even without a deep understanding of the subject I was able to follow along across the examples and empirical results. In particular, it was good to see the authors clearly lay out where their novel approach would work and where it would not and to be able to identify why in both cases.

My only real concern stems from the empirical results compared to some of the claims made early in the paper. Given the strength of the claims comparing the authors approach and prior approaches, it seems that the empirical results are somewhat weak. The authors make sure to put these results into context, but given the clarity of the results in the toy domains I would have expected clearer takeaways from the empirical results as well.

Edit: The authors greatly improved the paper, addressing all major reviewer concerns.

**Experience Assessment:**

I have read many papers in this area.

**Review Assessment: Checking Correctness Of Derivations And Theory:**

I did not assess the derivations or theory.

**Review Assessment: Checking Correctness Of Experiments:**

I assessed the sensibility of the experiments.

**Review Assessment: Thoroughness In Paper Reading:**

I made a quick assessment of this paper.

---

> ### Author Response · Authors · 2019-11-13
> **Response to Reviewer 3**
>
> We thank the reviewer for the positive and constructive feedback.
>
> Thank you for the suggestion regarding the empirical results. We clarify the takeaways from the paper below, and have worked this commentary into the most recent version of the paper:
> Validation of the feasibility of the PBR oracle in normal form games (NFGs): the asymmetric nature of these games, in combination with the number of players and strategies involved, makes them inherently, and perhaps surprisingly, large in scale. For example, our largest NFG involves 5 players with 30 strategies each, making for >24 million strategy profiles in total, which we note is well beyond the scale of canonical NFG domains. Overall, despite their stateless nature, we consider these NFG experiments as key empirical results, in contrast to the toy domains used in our counterexamples.
>
> Alpha-PSRO lowering NashConv in 2-player poker experiments: while Alpha-Rank does not seek to find an approximation of Nash, it nonetheless reduces the NashConv yielding extremely competitive results in comparison to an exact-Nash solver in these instances. This result is both non-obvious and quite important, in the sense of establishing Alpha-PSRO as a convenient means of training agents in >2-player games (where Nash is not readily computable).
>
> MuJoCo soccer experiments: Although noted as preliminary, a key observation can be made from these results: upon completion of training, when computing a new play distribution based on a pool of agents trained via the AlphaRank-based training approach vs. a uniform approach, the former attains essentially all of the ‘play probability’. This is evident in the colorbar on the far right of Appendix F, Fig. F.10, which visualizes the post-training meta-distribution over both training pipelines. Overall, we agree this insight should have been provided more clearly in the text.
>
> Based on your feedback, we have updated the revision to integrate changes related to the above discussions. Please let us know if further clarification of any of these points are needed.
>
> Finally, on note related to Reviewer 1 and 3’s feedback, we are investigating several additional experiments with the aim to include them in the revision before the author discussion period closes. (We will post an update as soon as applicable regarding any new results.)

---

> ### Author Response · Authors · 2019-11-15
> **Follow-up Response to Reviewer 3: Additional experiments**
>
> As promised, we have conducted new experiments on the MuJoCo soccer domain, which demonstrate the effectiveness of the alpha-PSRO training procedure against a self-play training procedure. Specifically, these give insights on performance improvements resulting from PSRO vs. standard population-based training regimes, in addition to the comparisons of PSRO meta-solvers evaluated in the original experiments.  Please see Appendix F and Fig. F.11 for these updated results.
>
> Additionally, we have now appended new Poker results to the latest revision. These include an evaluation of our training approach against the rectified Nash solver introduced in Balduzzi et al. (ICML, 2019) in two player games; we have additionally included rectified projected replicator dynamics as a comparison baseline to those experiments. Please see Fig. 3 for the updated experiments. We have likewise updated the text in Section 5 (“Evaluation”) to convey insights into these new results. Additionally, due to the rather counterintuitive nature of the rectified Nash experiments, we have appended a new section (Appendix C.5: “Explanation of Rectified Nash Performance”) with a walkthrough of the results.

---

### Official Review · AnonReviewer2 · 2019-10-22
**Official Blind Review #2**

**Rating:** 8

**Review:**

This paper extends the original PSRO paper to use an $\alpha$-Rank based metasolver instead of the projected replicator dynamics and Nash equilibria based metasolvers in the original. To this end, the paper modifies the original idea of Best-Response (BR) oracle since it can ignore some strategies in $\alpha$-Rank defining SSCC to introduce the idea of _preference-based_ Best-Response (PBR) oracle. The need for a different oracle is well justified especially with the visualization in the Appendix. The main contributions that the paper seems to be going for is a theoretical analysis of $\alpha$-Rank based PSRO compared to standard PSRO. From the PBR's description (especially in Sec 4.3) it seems the paper is intereseted in expanding the population with novel agents rather than finding the "best" single agent which is not well defined for complex games with intransitivities. Nevertheless, it seems that BR is mostly compatible with PBR for symmetric zero-sum two-player games.
The paper performs empirical experiments on different versions of poker. First set of experiments compare BR and PBR with $\alpha$-Rank based metasolver on random games and finds that PBR does better than BR at population expansion as defined. The second set of experiments compare the metasolvers. $\alpha$-Rank performs similarly to Nash where applicable. Moreover it's faster than Uniform (fictitious self-play) on Kuhn. Then the paper tacks on the MuJoCo soccer experiment as a teaser for ICLR crowd.

Overall the paper is quite interesting from the perspective of multiagent learning and I would lean towards accepting. However the paper needs to clarify a lot of details to have any chance of being reproducible.

** Clarifications needed:

- Tractability of PBR-Score and PCS-Score
It's unclear how tractable these are. Moreover these were only reported for random games. What did these scores look like for the Poker games? Could you clarify how exactly these were computed?

- It's somewhat unclear what the lack of convergence without novelty-bound oracle implies. Does this have to do with intransitivities in the game?

- Dependence of $\alpha$?
The original $\alpha$-Rank paper said a lot about the importance of choosing the right value for $\alpha$. How were these chosen? Do you do the sweep after every iteration of PSRO?

- Oracle in experiments?
The paper fails to mention the details about the Oracles being used in the experiments. They weren't RL oracles but more details would be useful.

- BR not compatible with PBR, albeit not the other way around, meaning one of the solutions you get from PBR might be BR, but can we say which one?

- For MuJoCo soccer was it true PSRO or cognitive hierarchy. In general, the original PSRO paper was partly talking about the scalable approach via DCH. This paper doesn't mention that at all. So were the MuJoCo experiments with plain PSRO? What was the exact protocol there? From the appendix it's unclear how the team-vs-team meta game works with individual RL agents. Moreover how are the meta-game evaluation matrices computed in general? How many samples were needed for the Poker games and MuJoCo soccer?

- The counterexamples in Appendix B3 are quite interesting. Do you have any hypotheses about the disjoint support from games' correlated equilibria?

**Experience Assessment:**

I have published one or two papers in this area.

**Review Assessment: Checking Correctness Of Derivations And Theory:**

I assessed the sensibility of the derivations and theory.

**Review Assessment: Checking Correctness Of Experiments:**

I carefully checked the experiments.

**Review Assessment: Thoroughness In Paper Reading:**

I read the paper thoroughly.

---

> ### Author Response · Authors · 2019-11-13
> **Response to Reviewer 2 [Part 1]**
>
> We thank the reviewer for the detailed feedback. We agree that clarifying these points is useful for reproducibility and also building reader intuition on the results. Please find our point-by-point responses below, which have been integrated into the latest revision.
>
> Tractability of PBR-Score and PCS-Score:
>
> This is an important and insightful question regarding the tractability of convergence measures such as PBR- and PCS-Scores. We developed these scores to assess the quality of convergence in our examples, in a manner analogous to NashConv. The computation of these scores is, however, not tractable in general games. Notably, this is also the case for NashConv (as it requires computation of player-wise best responses, which can be problematic even in moderately-sized games). Despite this, these scores remain a useful way to empirically verify the convergence characteristics in small games where they can be tractably computed.
>
> We agree that this is a useful remark for readers interested in implementing these scores, and have revised the paper to do so in Section C.3. Additionally, we now include pseudocode, in the same section, detailing how to compute these scores.
>
>
> Intuition on lack of convergence without novelty-bound oracle:
> As the reviewer points out, the lack of convergence without a novelty-bound oracle is precisely related to game intransitivities, i.e. cycles in the game can trap the oracle without the novelty-bound constraint. We show an example of this occurring in the revised paper Appendix B.4 (Figure B.7). Specifically, SSCCs may be hidden by “intermediate” strategies that, while not receiving as high a payoff as current population-pool members, can actually lead to well-performing strategies outside the population. As these “intermediate” strategies are avoided, SSCCs are consequently not found. Note also that this is related to the common problem of action/equilibrium shadowing (See Matignon et al., 2012, “Independent reinforcement learners in cooperative Markov games: a survey regarding coordination problems”).
>
> Note that per Reviewer 1 and 2’s feedback, we have made several improvements to the descriptions of the above example, specifically appending a paragraph following Proposition 4 to better explain this intuition, updating some of the proof text in Section B.4, and relabeling Fig B.7’s captions. We hope these changes make the intuition clearer.
>
> Dependence on $\alpha$ parameter:
> Thanks for pointing this out. Indeed, for all alpharank results, we run a sweep over alpha after each PSRO iteration (as recommended in the original alpharank paper). We have updated Section C.1 (Experimental Procedures) of the revised paper to clarify this. Overall, relative to the other modules of the training pipeline, we did not find this to be a computational constraint, especially for the larger (>2-player) games and when using a sparse representation and solver for computing the alpharank distribution.
>
> On a related note, we have also added more details on the hyperparameters used for the projected replicator dynamics meta-solver to Section C.1.
>
> Oracle in experiments:
> The oracles used in the experiments were (exact) best response oracles, computed by traversing the game tree. Specifically, we used OpenSpiel (https://github.com/deepmind/open_spiel) as the backend for the experiments using the exact best response oracle. Specifics of the implementation can be found in https://github.com/deepmind/open_spiel/blob/master/open_spiel/python/algorithms/best_response.py). We’ve updated Section C.1 (Experimental Procedures) to provide these details. Please let us know if this clarifies things. Many thanks!
>
> BR-PBR compatibility:
> We thank the reviewer for this question, as the concept of compatibility between objectives benefitted from a clarifying example. In general, BR and PBR optimize different objectives. However, in certain types of games (e.g., win-loss and monotonic games, defined respectively in Propositions 5 & 6), the strategy that maximizes value also maximizes the amount of other strategies beaten. In other words, this makes BR compatible with PBR, in the sense that the BR solution space is a subset of the PBR solution space.
>
> To make these properties clearer for readers, we have added an example comparing BR and PBR in a monotonic game in Figure B.8 of the appendix. In the case of Win-Loss games, PBR and BR optimize exactly the same objective, and therefore have the same solutions.

---

> > ### Author Response · Authors · 2019-11-13
> > **Response to reviewer 2 [Part 2]**
> >
> > MuJoCo soccer (true PSRO vs. cognitive hierarchy):
> >
> > The training approach we used in the experiment comparing PSRO-Alpharank with PSRO-Uniform corresponds to PSRO, rather than DCH, with each ‘PSRO step’ consisting of 1 billion training steps in the underlying game. Specifically, in the MuJoCo setting evaluated, each team was composed of several clones of a unique RL agent. Meta-game evaluations were conducted by composing a team of identical agents, and facing them off against a team of  other identical agents. E.g., in a 3-vs-3 game, a pool of 2 agents {A, B} would yield a 2x2 meta-payoff table with the following 4 entries: (AAA vs. AAA), (AAA vs. BBB), (BBB vs. AAA), (BBB vs. BBB). Effectively, the team-vs-team metagame is thus also the agent-vs-agent metagame, thereby enabling us to conduct our analysis on a matrix, instead of a tensor of rank (2 * team size).
> >
> > For the poker results, per iteration of PSRO, we used 100 simulations per entry of the meta-payoff table. For soccer experiments, the number of simulations per entry were adaptive to alleviate the cost of simulating this significantly more complex domain. An average of 10 to 100 simulations were conducted per entry, with fewer simulations used for meta-payoffs with higher certainty. Payoff uncertainties were estimated by computing the standard deviation of a beta-law of parameter (matches won, matches lost). For the final evaluation matrix reported in Appendix F (Fig. F.10), which was computed after the conclusion of PSRO-based training, 100 simulations were used per entry.
> >
> > We have updated Sections C.1 and F of the revised paper appendix to include these details.
> >
> > Counterexamples in Appendix B3:
> > [Please note that Appendix B.3 is now A.2, due to updates in the revised paper.]
> >
> > This is a great question. Indeed, the notion of strategy defection underlies both alpharank and correlated equilibria, although in quite different ways; alpharank is motivated by evolutionary dynamics and is built off the notion of unilateral defection from individual strategy profiles, whereas correlated equilibria are defined in terms of defections from distributions over profiles. We expect that these differences (in the manner in which the two solution concepts use the notion of defection) could be used to pinpoint the precise relations between the two, although leave this for future work.

---

> > > ### Comment · AnonReviewer2 · 2019-11-14
> > > **Re:**
> > >
> > > I'm impressed with the response in the rebuttal and looking forward to the updates to your experiments. Meanwhile I have updated my score. Given the interest in applying to continuous action problems, would be useful to cite other related works in the literature like:
> > >
> > > Iqbal, S., & Sha, F. (2019, May). Actor-Attention-Critic for Multi-Agent Reinforcement Learning. In International Conference on Machine Learning (pp. 2961-2970).
> > >
> > > Gupta, J. K., Egorov, M., & Kochenderfer, M. (2017, May). Cooperative multi-agent control using deep reinforcement learning. In International Conference on Autonomous Agents and Multiagent Systems (pp. 66-83). Springer, Cham.
> > >
> > > Wei, E., Wicke, D., Freelan, D., & Luke, S. (2018, March). Multiagent soft q-learning. In 2018 AAAI Spring Symposium Series.

---

> > > > ### Author Response · Authors · 2019-11-15
> > > > **Thanks !**
> > > >
> > > > We appreciate your constructive feedback, which definitely helped to improve the paper’s quality. Thanks also for your kind remarks and for updating the score (although, it appears to not have changed on our end, though maybe it becomes visible after the review process? We wanted to kindly flag this just in case. Thanks again!)
> > > >
> > > > Regarding the new experiments, please see our response to your other comment for details.
> > > >
> > > > Thanks also for the suggestion on the additional works to include, which have all been appended to the related works in the latest revision. Additionally, we have added the following recent related works for interested readers:
> > > >
> > > > Hernandez-Leal, Pablo, Bilal Kartal, and Matthew E. Taylor. "A survey and critique of multiagent deep reinforcement learning." Autonomous Agents and Multi-Agent Systems (2019): 1-48.
> > > >
> > > > Khadka, Shauharda, Somdeb Majumdar, and Kagan Tumer. "Evolutionary Reinforcement Learning for Sample-Efficient Multiagent Coordination." arXiv preprint arXiv:1906.07315 (2019).
> > > >
> > > > Peng, Peng, et al. "Multiagent bidirectionally-coordinated nets: Emergence of human-level coordination in learning to play starcraft combat games." arXiv preprint arXiv:1703.10069 (2017).

---

> > > ### Comment · AnonReviewer2 · 2019-11-14
> > > **MuJoCo**
> > >
> > > When you say team of identical agents (clones), you mean they share the exact same weights? This seems more like the setting in [1] with homogeneous agents?
> > > Moreover, didn't [2] have two players in the team (2v2)? It seems here you have 3v3? If so I missed this detail from the paper.
> > >
> > > [1] Gupta, J. K., Egorov, M., & Kochenderfer, M. (2017, May). Cooperative multi-agent control using deep reinforcement learning. In International Conference on Autonomous Agents and Multiagent Systems (pp. 66-83). Springer, Cham.
> > >
> > > [2] Liu, S., Lever, G., Merel, J., Tunyasuvunakool, S., Heess, N., & Graepel, T. (2018). Emergent Coordination Through Competition.

---

> > > > ### Author Response · Authors · 2019-11-15
> > > > **Followup**
> > > >
> > > > That’s correct, the clones on each team have the exact same weights (i.e., homogeneous teams as in [1] (Gupta 2017)). We now make this difference with respect to the heterogenous teams evaluated in [2] (Liu 2018) clear in the revised paper (Section F). As the primary objective of our paper was to analyze (empirically and theoretically) PSRO’s performance with novel meta-solvers and in >2-player games, the MuJoCo experiments serve as a preliminary evaluation of the scalability of the approach to more complex domains. We completely agree that investigation of heterogeneous teams (as in [2]) is also interesting, particularly from a behavioral diversity perspective, though leave this for future work.
> > > >
> > > > As promised, we have conducted new experiments on the MuJoCo soccer domain, which demonstrate the effectiveness of the alpha-PSRO training procedure against a self-play training procedure. Specifically, these give insights on performance improvements resulting from PSRO vs. standard population-based training procedures, in addition to the comparisons of PSRO meta-solvers evaluated in the original experiments.  Please see Appendix F and Fig. F.11 for these updated results. Indeed our earlier results were for 3v3 teams. We evaluate on 2v2 teams in these new experiments, to bear a closer similarity to [2].
> > > >
> > > > Additionally, we have now appended new Poker results to the latest revision. These include an evaluation of our training approach against the rectified Nash solver introduced in Balduzzi et al. (ICML, 2019) in two player games; we have additionally included rectified projected replicator dynamics as a comparison baseline to those experiments. Please see Fig. 3 for the updated experiments. We have likewise updated the text in Section 5 (“Evaluation”) to convey insights into these new results. Additionally, due to the rather counterintuitive nature of the rectified Nash experiments, we have appended a new section (Appendix C.5: “Explanation of Rectified Nash Performance”) with a walkthrough of the results.

---

### Official Review · AnonReviewer1 · 2019-10-23
**Official Blind Review #1**

**Rating:** 8

**Review:**

Review Update (18/11/2019)
Thank you for the detailed replies and significant updates to the paper in response to all reviewers. You have comfortably addressed all of my concerns and so I have updated my score. I think the paper has improved significantly through the rebuttal stage and therefore the update in my score is also significant to match the far larger contribution to the community that the paper now represents.

--
This paper considers alpha-rank as a solution concept for multi-agent reinforcement learning with a focus on its use as a meta-solver for PSRO. Based on theoretical findings showing shortcomings of using the typical best response oracle, the paper finds a necessity for a new response oracle and proposes preference-based best response.

The theoretical contributions help further the community's understanding of alpha-rank but the method remains somewhat disconnected from other recent related literature. Therefore, I think the paper's subsequent impact could be significantly improved by making more direct comparison to recent results. Specifically:

1) In the 2-player games comparisons are currently made to PRD based on its use in Lanctot et al (NeurIPS, 2017) instead of the more recent PSRO Rectified Nash approach proposed by Balduzzi et al. (ICML, 2019). Please make this direct comparison or justify its exclusion.

2) The preliminary MuJoCo soccer results in Appendix G significantly increase the relevance of this work to the ICLR community given the prior publication of this environment at ICLR 2019. However, the results are currently incomplete. In particular, to again strengthen the link to existing work, comparison of the method proposed in this paper to the agents trained by population based training in Liu et al. (ICLR, 2019) would be a more informative comparison than the preliminary results presented in comparison to the naïve uniform meta-solver.

3) Appendix A includes a brief literature survey. This is important material to position the paper in relation to existing work, particularly for readers not familiar with the area that will rely on this to understand the paper as a self contained reference. Please move this section into the main body of the paper and expand to fully credit the work this paper builds upon.


Minor Comments:
In Appendix C.4 should the reference to Figure C.7 be to Figure C.7a specifically? and the reference to Figure C. 7a be to Figure C. 7b-f inclusive? If so, I believe the available joint strategies in step 4 is missing (1,1,2) as shown in Figure C. 7f.


**Experience Assessment:**

I have published in this field for several years.

**Review Assessment: Checking Correctness Of Derivations And Theory:**

I assessed the sensibility of the derivations and theory.

**Review Assessment: Checking Correctness Of Experiments:**

I carefully checked the experiments.

**Review Assessment: Thoroughness In Paper Reading:**

I read the paper thoroughly.

---

> ### Author Response · Authors · 2019-11-13
> **Response to Reviewer 1**
>
> We thank the reviewer for the detailed feedback, which we address below. We are currently integrating this feedback into the revision.
>
> Main feedback:
> We are currently running several additional experiments related to those suggested, with the aim to update the paper before the author discussion period closes. We will post an update as soon as new results are available.
>
> We completely agree regarding the related works section, and have moved it back to the main body in the latest revision (Sec. 6).
>
> Minor comments:
> Thanks for pointing out the issue with figure references, which have been corrected in the revision as you specified (please note that the referenced section and figure are now, respectively, Appendix B.4 and Fig. B.7, due to the related works section being moved out of the appendix). We’ve also updated the subfigure captions to make the correspondence to the counterexample steps clear. Indeed the strategy space in Step 4 should have included (1,1,2) — thanks for catching this!

---

> ### Author Response · Authors · 2019-11-15
> **Followup Response to Reviewer 1**
>
> As promised, we have conducted new experiments on the MuJoCo soccer domain, which demonstrate the effectiveness of the alpha-PSRO training procedure against a self-play training procedure. Specifically, these give insights on performance improvements resulting from PSRO vs. standard population-based training pipelines, in addition to the comparisons of PSRO meta-solvers evaluated in the original experiments.  Please see Appendix F and Fig. F.11 for these updated results. Given the differences between our training procedure and that of Liu et al. (ICLR, 2019) and review period timelines, this was the closest we could come to comparing the differences between PSRO-based opponent sampling and the PBT-style opponent sampling method used in Liu et al. (ICLR, 2019), while keeping all other aspects of our method fixed.
>
> Additionally, we have now appended new Poker results to the latest revision. These include an evaluation of our training approach against the rectified Nash solver introduced in Balduzzi et al. (ICML, 2019) in two player games; we have additionally included rectified projected replicator dynamics as a comparison baseline to those experiments. Please see Fig. 3 for the updated experiments. We have likewise updated the text in Section 5 (“Evaluation”) to convey insights into these new results. Additionally, due to the rather counterintuitive nature of the rectified Nash experiments, we have appended a new section (Appendix C.5: “Explanation of Rectified Nash Performance”) with a walkthrough of the results.

---

### Decision · Program_Chairs · 2019-12-19

**Decision:**

Accept (Talk)

**Comment:**

This paper analyzes and extends learning methods based on Policy-Spaced Response Oracles (PSRO) through the application of alpha-rank.  In doing so, the paper explores connections with Nash equilibria, establishes convergence guarantees in multiple settings, and presents promising empirical results on (among other things) 3-to-5 player poker games.

Although this paper originally received mixed scores, after the rebuttal period all reviewers converged to a consensus. A revised version also includes new experiments from the MuJoCo soccer domain, and new poker results as well.  Overall, this paper provides a nice balance of theoretical support and practical relevance that should be of high impact to the RL community.